# Supervised Guidance Training for Infinite-Dimensional Diffusion Models

**Elizabeth L. Baker** [* 1]   **Alexander Denker** [† * 2]   **Jes Frellsen** [1]

## Abstract

Score-based diffusion models have recently been extended to infinite-dimensional function spaces, with uses such as inverse problems arising from partial differential equations. In the Bayesian formulation of inverse problems, the aim is to sample from a posterior distribution over functions obtained by conditioning a prior on noisy observations. While diffusion models provide expressive priors in function space, the theory of conditioning them to sample from the posterior remains open. We address this, assuming that either the prior lies in the Cameron-Martin space, or is absolutely continuous with respect to a Gaussian measure. We prove that the models can be conditioned using an infinite-dimensional extension of Doob's $h$-transform, and that the conditional score decomposes into an unconditional score and a guidance term. As the guidance term is intractable, we propose a simulation-free score matching objective (called *Supervised Guidance Training*) enabling efficient and stable posterior sampling. We illustrate the theory with numerical examples on Bayesian inverse problems in function spaces. In summary, our work offers the first function-space method for fine-tuning trained diffusion models to accurately sample from a posterior.

## 1. Introduction

Inverse problems, which aim to infer unknown parameters from observations, are ubiquitous in science and engineering. These problems are typically ill-posed, meaning that the solution may be unstable in the presence of noise, and multiple parameters may map to the same observed state. The Bayesian framework offers a principled solution by targeting a posterior distribution conditioned on data rather than a single point estimate (Stuart, 2010). Frequently, these problems originate from partial differential equations (PDEs), where the unknown parameters exist in function spaces. This means that the target probability measures are inherently infinite-dimensional.

While score-based diffusion models (SDMs) have recently been extended to infinite dimensions (Franzese et al., 2023; Pidstrigach et al., 2024; Lim et al., 2025; Franzese & Michiardi, 2025), conditioning such models to sample from posterior distributions remains an open problem. In finite-dimensional settings, conditioning is often achieved via Doob's $h$-transform (Rogers & Williams, 2000; Zhao et al., 2025), where the conditional score decomposes into an unconditional score and a guidance term (Dhariwal & Nichol, 2021; Chung et al., 2023). Various methods have been proposed to approximate this typically intractable guidance term, ranging from heuristic approximations (Chung et al., 2023; Yu et al., 2023; Finzi et al., 2023) to learning the term directly (Denker et al., 2024; Zhang et al., 2023). Some of these methods have been extended to infinite-dimensional SDMs. Baldassari et al. (2023) propose a framework to train a conditional infinite-dimensional SDM to directly estimate the posterior distribution. FunDPS (Yao et al., 2025) extends DPS (Chung et al., 2023) to infinite dimensions in the case where the data is contained in the Cameron-Martin space of the Wiener process.

Lifting these results into infinite dimensions is not straightforward. For example, Lebesgue densities do not exist, so score functions must be designed without referring to densities. Furthermore, the definition of conditional scores relies on transition operators of the forward and reverse stochastic differential equations (SDEs) being differentiable; this does not always hold in infinite dimensions. We must also contend with issues arising from the Feldman-Hájek theorem: shifting Gaussian measures by an element not in their Cameron-Martin space leads to mutually singular measures. This means we cannot rely on the existence of densities between the data and the measure induced by the Wiener process of the SDE.

At the same time, if we ignore the infinite-dimensional setting by discretising the data immediately, it is unclear how good the approximation is and whether the solutions will be-

---

[*]Equal contribution [†]Work done while at University College London. [1]Department of Applied Mathematics and Computer Science, Technical University of Denmark, Denmark [2]Helmholtz Imaging, Deutsches Elektronen-Synchrotron DESY, Germany. Correspondence to: Elizabeth Baker <eloba@dtu.dk>.

*Proceedings of the 43rd International Conference on Machine Learning*, Seoul, South Korea. PMLR 306, 2026. Copyright 2026 by the author(s).

have as the resolution of the discretisation is increased. Prior work has shown applying finite-dimensional architectures to higher resolutions often degrades in performance, see the discussion in Appendix C.4. By treating the theory in infinite dimensions and only discretising in the last step, we can better understand the errors induced by discretisation. Moreover, we unlock alternative and more flexible options in terms of discretisation, training and loss functions. This results in a more general resolution-invariant framework for conditioning infinite-dimensional SDMs.

**Contributions** In this work, we develop a principled framework for conditioning diffusion models in function space using an infinite-dimensional version of Doob's $h$-transform. We prove the existence of the conditional diffusion process under two general settings: (1) the prior distribution lies in the Cameron-Martin space of the Wiener process used in the diffusion process, or (2) the prior is absolutely continuous with respect to a Gaussian measure. Further, we prove a decomposition of the conditional score term into the unconditional score term and a guidance term.

We introduce *Supervised Guidance Training*, an algorithm for learning the typically intractable guidance term. This approach is based on a denoising score matching loss and thus avoids the computational overhead of simulation-based training regimes. Further, we explore connections between our framework, stochastic optimal control, and Tweedie-based approximations, where we prove the approximation in FunDPS (Yao et al., 2025) with less restrictive assumptions. Finally, we validate our method on Bayesian inverse problems, demonstrating that explicitly learning the infinite-dimensional guidance term yields superior accuracy compared to heuristic baselines.

# 2. Background and Setup

## 2.1. Bayesian Inverse Problems

A major motivation for our work is Bayesian inverse problems in function spaces (Stuart, 2010; Dashti & Stuart, 2017). Consider a measurable forward operator $G : \mathcal{H} \to \mathcal{Y}$ for two Hilbert spaces $\mathcal{H}$ and $\mathcal{Y}$ and assume that we have access to observations of the form $y = G(f) + \eta$, where $f$ is the parameter to be identified and $\eta \in \mathcal{Y}$ denotes observation noise. We place a prior measure $\pi$ on $f \in \mathcal{H}$ and model the observation noise with a measure $\pi_0$ on $\mathcal{Y}$ such that $f$ and $\eta$ are independent. The conditional random variable $y|f$ then has a measure $\pi_f$, which is given as $\pi_0$ translated by $G(f)$. We assume $\pi_f \ll \pi_0$ for $f$ $\pi$-almost surely and for some potential $\Phi : \mathcal{H} \times \mathcal{Y} \to \mathbb{R}$, write the likelihood as

$$\frac{\mathrm{d}\pi^f}{\mathrm{d}\pi_0}(y) = \exp(-\Phi(f, y)). \tag{1}$$

By the infinite-dimensional Bayes theorem (Theorem 14 in Dashti & Stuart (2017)), the posterior satisfies $\pi^y \ll \pi$ for almost every $y$ and for $\xi = \mathbb{E}_{f \sim \pi}[\exp(-\Phi(f, y))]$

$$\frac{\mathrm{d}\pi^y}{\mathrm{d}\pi}(f) = \frac{1}{\xi} \exp(-\Phi(f, y)). \tag{2}$$

This framework supports Bayesian inference for PDE-based inverse problems in their natural function-space setting.

In many applications, the noise distribution can be selected or calibrated from prior experimental data, while the prior must be modelled to encode structural information about $f$. In this work, we construct this prior using an infinite-dimensional diffusion model, enabling a data-driven representation of the underlying function space.

## 2.2. Conditioning in Finite Dimensions

We start by conditioning SDMs in the finite-dimensional setting, with state space $\mathbb{R}^d$. Let $x_t$ be the forward process, satisfying an SDE with drift $\mathbf{f}(t, x)$ and diffusion $g(t)$, such that $x_0 \sim \pi_{\text{data}}$. Denote by $p_t(x)$ the marginal density of $x_t$. Letting $\tilde{t} := T - t$ and $\mathbf{w}_t$ be a Wiener process, the time-reversal of the forward SDE satisfies

$$\mathrm{d}z_t = \left[\mathbf{f}(\tilde{t}, z_t) + g^2(\tilde{t})\nabla_z \log p_{\tilde{t}}(z_t)\right] \mathrm{d}t + g(\tilde{t})\mathrm{d}\mathbf{w}_t, \tag{3}$$

with $z_0 \sim \text{Law}(x_T) \approx \mathcal{N}(0, C_{\pi_{\text{data}}})$, see Song et al. (2021). To solve an inverse problem given observations $y$, we seek to sample from the posterior distribution $p_0(x_0|y)$. This requires conditioning the reverse process $z_t$ on $y$, which can be achieved using Doob's $h$-transform (Rogers & Williams, 2000). In particular, conditioning $z_t$ is equivalent to adding a *guidance term* $\nabla_z \log h(\tilde{t}, z_t)$, scaled by $g(\tilde{t})^2$, to the drift of $z_t$, which forces the trajectory towards the observations $y$. In finite dimensions, the optimal guidance is defined with

$$h(t, x_t) \propto \mathbb{E}[p(y \mid x_0) \mid x_t] = p(y \mid x_t). \tag{4}$$

Using Bayes' rule, the guidance decomposes into two terms,

$$\nabla_x \log p(y \mid x_t) = \nabla_x \log p(x_t \mid y) - \nabla_x \log p_t(x_t). \tag{5}$$

The first term corresponds to the conditional score, whereas the second term is learned by the unconditional diffusion model. Many approaches aim to approximate the expectation in (4) and thus the guidance term (Chung et al., 2023; Rout et al., 2024). As an alternative, some approaches learn the guidance term (Denker et al., 2024) or the conditional score directly (Batzolis et al., 2021). Extending these concepts to function spaces requires careful treatment of the underlying measures, as Lebesgue densities do not exist and the score decomposition in (5) must be justified via the $h$-transform in Hilbert spaces.

## 2.3. Infinite-dimensional Score-Based Diffusion Models

Various formulations for infinite-dimensional diffusion models have been explored, see e.g. Franzese et al. (2023); Lim et al. (2025); Hagemann et al. (2025); Pieper-Sethmacher & Paulin (2026) or Pidstrigach et al. (2024). We follow Pidstrigach et al. (2024), which allows for a general setup, including the case where the data distribution is not supported on the Cameron-Martin space of the Wiener process. Suppose we have a data distribution $\pi$ that is supported on a Hilbert space $\mathcal{H}$. Let $C$ be a trace-class covariance operator and $W^{\mathcal{H}}$ the cylindrical Wiener process associated to $\mathcal{H}$. Then we define an infinite-dimensional forward SDE starting from the prior data distribution $\pi$

$$dX_t = -\frac{1}{2}X_t dt + \sqrt{C}dW_t^{\mathcal{H}}, \quad X_0 \sim \pi, \qquad (6)$$

for which the marginal distribution of $X_t$ will converge to the stationary distribution $\mathcal{N}(0, C)$ as $t \to \infty$.

*Remark* 2.1. For notational ease, we will consider SDEs of the form (6). However, the arguments in this paper and by Pidstrigach et al. (2024) also apply to the commonly used variance-preserving SDE formulation.

In this setting, the time-reversal $Z_t := X_{T-t}$ satisfies

$$dZ_t = \left[\frac{1}{2}Z_t + s(T - t, Z_t)\right]dt + \sqrt{C}dW_t^{\mathcal{H}}, \quad (7)$$

where $Z_0$ starts from noise, $Z_0 \sim \text{Law}(X_T) \approx \mathcal{N}(0, C)$, and the score $s(t, x)$ is defined as

$$s(t, x) := -\alpha_t \left(x - e^{-\frac{t}{2}}\mathbb{E}[X_0 | X_t = x]\right), \qquad (8)$$

with $\alpha_t := (1 - e^{-t})^{-1}$. In finite dimensions, this is equivalent to the standard score-based diffusion models, with $s(t, x) = \nabla_x \log p_t(x)$ (see (3)).

We consider the the infinite-dimensional process in (7) as the unconditional SDE, which samples from a prior $\pi$. The goal of this paper is to show that, under suitable assumptions, (7) can be conditioned to sample from the posterior distribution $\pi^y$.

Recently, Baldassari et al. (2023) studied the conditional sampling problem but with a different approach. Rather than assuming access to the unconditional score $s(t, x)$, they directly define a conditional score

$$s^y(t, x) := -\alpha_t \left(x - e^{-\frac{t}{2}}\mathbb{E}[X_0 | X_t = x, Y = y]\right), \quad (9)$$

where as before, we use $\alpha_t := (1 - e^{-t})^{-1}$. They show that replacing $s(t, x)$ by $s^y(t, x)$ in (7) leads to a conditional reverse SDE that samples from $\pi^y$ and establish conditions under which this SDE is well-posed.

## 3. Conditioning via Doob's $h$-Transform

Instead of learning the conditional score directly, we instead aim to condition the reverse SDE (7). Similar to the finite-dimensional framework, we prove that we can do this by adding an additional term, given by Doob's $h$-transform, to the drift of the reverse SDE. We prove this under two commonly used settings:

**Setting 1.** The support of the prior $\pi$ is contained in a ball of radius $R$ in the Cameron-Martin space of $C$, with $C$ the covariance of the diffusion model SDE in (6);

**Setting 2.** The prior is absolutely continuous with respect to a Gaussian distribution, $\frac{d\pi}{d\mathcal{N}(0, C_\pi)}(u) = \frac{1}{\chi}\exp(-\Psi(u))$, for some covariance operator $C_\pi$. Moreover, we ask that $E_0 \leq \Psi(u) \leq E_1 + E_2\|u\|^2$ and that $\nabla\Psi$ is Lipschitz continuous. We also make the assumption that $\Phi$ is also differentiable with $\nabla\Phi$ continuous.

The first setting is studied by Yao et al. (2025), whereas the second is the main setting of Baldassari et al. (2023). For both settings, the time-reversed SDE has a strong solution, by Theorem 12 and Theorem 13 by Pidstrigach et al. (2024).

To illuminate these two settings, suppose the prior is Gaussian, $\pi = \mathcal{N}(0, C_\pi)$. Then the support of $\mathcal{N}(0, C_\pi)$ is not contained in the Cameron-Martin space of $C_\pi$. For Setting 1 we would thus need to choose $C$ such that its Cameron-Martin space is large enough to contain the support of $\mathcal{N}(0, C_\pi)$. Alternatively, we can work in Setting 2, and set $C = C_\pi$ in the diffusion model (6). Since the distance to the target measure $\pi$ is given partly by the Wasserstein-2-distance of $\mathcal{N}(0, C)$ to $\pi$ (Pidstrigach et al., 2024), choosing $C = C_\pi$ will minimise the approximation error. On the other hand, if we do not have much information about the form of $\pi$ such as whether it has density with respect to a Gaussian, then indeed it makes sense to use Setting 1, and just ensure that the Cameron-Martin space of $C$ is big enough to contain the support of $\pi$.

We will be using an infinite-dimensional Doob's $h$-transform (Baker et al., 2024; Pieper-Sethmacher et al., 2025) to condition the time-reversed SDE in (7). For this, we identify an appropriate function $h$ and then show that this is the *correct* transform, in that if we sample from the conditioned SDE we sample from $\pi^y$. Finally, we show that the resulting score decomposes into the (already) learned score and a guidance term. With this setup, we define the $h$-transform

$$\boxed{h^y(t, x) := \xi^{-1}\mathbb{E}_{\mathbb{P}}[\exp(-\Phi(X_0, y)) \mid X_t = x],} \quad (10)$$

for $\xi = \mathbb{E}_{u \sim \mathbb{P}}[\exp(-\Phi(u, y))]$ and $\Phi$ the potential from (1).

**Theorem 3.1.** *Let $\mathbb{P}$ be the path measure of the unconditional time-reversal $Z_t$ in (7), with $h^y$ as in (10). We assume either Setting 1 or Setting 2. Then the regular conditional*

probability $\mathbb{P}(\cdot \mid Y = y)$, denoted as $\mathbb{P}^y$, satisfies

$$d\mathbb{P}^y = h^y(T, Z_T)d\mathbb{P}. \qquad (11)$$

Moreover, the conditional process of $Z_t$ given $Y = y$, which we denote by $Z_t^y$, satisfies

$$\mathrm{d}Z_t^y = b(t, Z_t^y)\mathrm{d}t + \sqrt{C}\mathrm{d}W_t, \quad Z_0^y \sim Law(X_T), \qquad (12)$$

$$b(t, z) := \frac{1}{2}z + s(T - t, z) + C\nabla \log h^y(T - t, z), \qquad (13)$$

where $\nabla \log h^y(t, z)$ is the Riesz representation of the Fréchet derivative of the logarithm of $h$ with respect to $z$.

*Proof.* The idea is to apply the results of the infinite-dimensional Doob's $h$-transform in Baker et al. (2024). The main result to show is that $h^y$ is differentiable. In Setting 2 this comes from differentiability of $\Phi$, whereas in Setting 1, this comes from an application of the Cameron-Martin theorem. See Section A.1 for details. $\qquad\square$

Theorem 3.1 shows that $h^y$ in (10) is the correct transform for sampling from the posterior $\pi^y$. In the finite-dimensional framework, we can split the posterior score into two terms: one learned by the unconditional score model, and the second a guidance term, see (5). In the next Theorem, we show that this decomposition also holds in infinite dimensions. In particular, the conditional score from Baldassari et al. (2023) can be written as $s^y(t, x) = s(t, x) + C\nabla \log h^y(t, x)$.

**Theorem 3.2.** *Assume either Setting 1 or Setting 2. Then, setting $\beta_t = \frac{e^{-\frac{t}{2}}}{1 - e^{-t}}$, we may decompose $C\nabla \log h(t, x)$ as*

$$C\nabla \log h^y(t, x) = s^y(t, x) - s(t, x) \qquad (14)$$

$$= \beta_t \left( \mathbb{E}[X_0 \mid X_t = x, Y = y] - \mathbb{E}[X_0 \mid X_t = x] \right). \qquad (15)$$

*Therefore, letting $Z_0^y \sim Law(X_T)$, then $Z_t^y$ satisfies*

$$\mathrm{d}Z_t^y = \frac{1}{2}Z_t^y\mathrm{d}t + s^y(T - t, Z_t^y)\mathrm{d}t + \sqrt{C}\mathrm{d}W_t^H. \qquad (16)$$

For a proof see Section A.2. These two Theorems show that in order to condition the reverse SDE (7), we require access to the *guidance term* $C\nabla \log h^y(t, x)$. However, as the $h$-transform (10) is intractable, due to its dependence on the full transition law of the infinite-dimensional process, it is not readily applicable and we require approximations.

# 4. Approximating the Guidance Term

We will study different ways of estimating and approximating the guidance term $C\nabla \log h^y$. First, we discuss the known connection to stochastic optimal control (SOC), which leads to a simulation-based loss function. Then, we derive a training-free approximation based on an infinite-dimensional Tweedie approximation. Finally, we will discuss our simulation-free training framework in Section 4.3.

## 4.1. Stochastic Optimal Control

One can make use of SOC (Fabbri et al., 2017) to describe the guidance term, see e.g. Park et al. (2024) for diffusion bridges in function spaces. For this, let $\mathbb{Q}^u$ be the path measure induced by

$$dZ_t^u = (b(t, Z_t^u) + Cu(t, Z_t^u))dt + \sqrt{C}dW_t^{\mathbb{Q}^u}, \qquad (17)$$

where $b(t, z)$ is the drift of the reverse SDE (7). We have that $\mathbb{Q}^{u^*}$ is equal to the conditional path measure $\mathbb{P}^y$ if $u^*$ minimises the following SOC loss

$$\mathcal{J}(u) = \mathbb{E}_{\mathbb{Q}^u} \left[ \Phi(Z_T^u, y) + \frac{1}{2} \int_0^T \|\sqrt{C}u_t\|_{\mathcal{H}}^2 dt \right], \qquad (18)$$

see Appendix B for details. We can actually use $\mathcal{J}(u)$ as an objective to learn the guidance term. Here, the main advantage is that $\mathcal{J}(u)$ does not rely on ground truth data and only requires access to the terminal cost $\Phi$. However, the objective is inherently simulation-based. Evaluating the loss requires sampling trajectories from the path measure $\mathbb{Q}^u$ induced by the current control. Consequently, the gradient with respect to the control must be estimated through the sampling process, typically via path-wise estimators such as REINFORCE (Williams, 1992). These estimators can have a high variance and are particularly sensitive to the terminal cost $\Phi$. In early stages of training, when the control is poorly initialised, sampled trajectories may fail to reach informative regions of $\Phi$, which can result in flat or noisy gradient signals. For finite-dimensional diffusion models alternatives objectives have been proposed, see, e.g. (Domingo-Enrich et al., 2025; Denker et al., 2024; Han et al., 2025; Uehara et al., 2024; Pidstrigach et al., 2025), but we are not aware of any extensions to infinite dimensions.

## 4.2. Infinite-Dimensional Tweedie Approximation

Many finite-dimensional posterior sampling methods rely on heuristic approximation to the guidance term, see e.g. DPS (Chung et al., 2023). Recently, FunDPS (Yao et al., 2025) extended DPS to infinite-dimensional diffusion models. However, they only work in the more restrictive Setting 1. Here, we show that it also holds in Setting 2, where we only require absolute continuity with respect to a Gaussian measure. This generalises the applicability of FunDPS to a broader class of infinite-dimensional diffusion models.

Central to the approximation is the ability to estimate clean data $X_0$ from the noisy state $X_t$. In finite dimensions, Tweedie's formula provides a link between the score function and the posterior mean (Vincent, 2011; Efron, 2011). Based on the definition of the infinite-dimensional score in (8), an analogue holds in our Hilbert space setting.

**Proposition 4.1** (Infinite-dimensional Tweedie Estimate)**.** *Let $s(t, x)$ be the score function defined in (8). The condi-*

**Algorithm 1** Supervised Guidance Training (SGT)

**Require:** Covariance operator $C$, pre-trained score model $s_\theta(t, z)$, loss weighting, training data $\{X^{(i)}, Y^{(i)}\}_{i=1}^N$, batch size $B$, guidance function $u_\phi$
1: **while** Metrics not good enough **do**
2:     Subsample $\{x_0^{(i)}, y^{(i)}\}_{i=1}^B$ from $\{X^{(i)}, Y^{(i)}\}_{i=1}^N$
3:     $\epsilon^{(i)} \sim \mathcal{N}(0, I)$
4:     $t_i \sim \mathcal{U}([0, T])$
5:     $\sigma_{t_i} = \sqrt{1 - e^{-t_i}}$
6:     $x_t^{(i)} = e^{-1/2 t_i} x_0^{(i)} + \sigma_{t_i} C^{1/2} \epsilon^{(i)}$
7:     $\hat{s}_\theta^{(i)} = \texttt{stopgrad}(s_\theta(x_t^{(i)}, t_i))$
8:     $\hat{u}_\phi^{(i)} = u_\phi(x_t^{(i)}, y^{(i)}, t_i)$
9:     $r^{(i)} = (\hat{s}_\theta^{(i)} + \hat{u}_\phi^{(i)}) + \sigma_{t_i}^{-1} C^{1/2} \epsilon^{(i)}$
10:     $L(\phi) = \sum_{i=1}^B \|C^{-1/2} r^{(i)}\|_2^2$ {evaluate loss (23)}
11:     Gradient step with $\nabla_\phi L(\phi)$
12: **end while**

tional expectation of the initial state given the noisy state at time $t$, is given exactly by

$$\mathbb{E}[X_0 \mid X_t = x] = e^{\frac{t}{2}} \left( x + (1 - e^{-t}) s(t, x) \right). \quad (19)$$

*We denote this via* $\hat{x}_t(x) := \mathbb{E}[X_0 \mid X_t = x]$.

The result follows immediately from the definition of the score term $s(t, x)$ in (8). Note, using the time-reversal, we can equivalently write $\hat{x}_t(x) = \mathbb{E}[Z_T \mid Z_{T-t} = x]$.

We can derive an approximation as follows

$$C\nabla \log h^y(t, z) = C\nabla_z \log \mathbb{E}_\mathbb{P}[\exp(-\Phi(X_0, y)) \mid X_t = z]$$
$$\approx -C\nabla_z \Phi(\hat{x}_t(z), y),$$

where we replace the conditional expectation with the Tweedie estimate $\hat{x}_t(x)$. Taking the Fréchet derivative with respect to $z$, and applying the chain rule, we obtain the approximate guidance term for the score $u^y(t, z) = C\nabla_z \log h^y(t, z)$ with the additional scaling $\gamma > 0$,

$$u^y(t, z) \approx -\gamma C \left( \nabla_z \hat{x}_t(z) \right)^* \nabla_f \Phi(f, y) \big|_{f = \hat{x}_t(z)}. \quad (20)$$

While (20) enables approximate conditional sampling, it introduces significant computational overhead. A key drawback of the DPS-style approximation is that evaluating the Tweedie estimate in (19) requires access to the score function. Consequently, as the guidance term in (20) involves the Fréchet derivative of the Tweedie map, it necessitates backpropagation through the score model at every discretisation step of the SDE.

### 4.3. Learning the Guidance Term

In this section, we show how the guidance term can be learned using a simulation-free training framework. In par-

ticular, we start by defining a score matching loss, minimised over functions $u$, where the expectation is taken over the uniform distribution $t \sim \mathcal{U}[0, T]$, and the joint law $X_t, Y \sim \text{Law}(X_t, Y)$

$$\mathbb{E}[\|u(t, X_t, Y) - C\nabla \log h^Y(t, X_t)\|_K^2], \quad (21)$$

where $K$ is a Hilbert space, the choice of which we discuss in Section 5.1. Even though we have the representation $C\nabla \log h^y(t, x) = s^y(t, x) - s(t, x)$ and assume full knowledge of $s$, (21) is intractable, as we do not know $s^y(t, x)$.

We note we could learn $C\nabla \log h^y(t, X_t)$ directly, without relying on knowledge of the unconditional score $s(t, X_t)$ via a simulation based loss. This is done for non-linear stochastic differential equations in finite dimensions by Heng et al. (2025); Baker et al. (2025); Pidstrigach et al. (2025); Yang et al. (2025b), and also considered in infinite dimensions in Baker et al. (2024); Yang et al. (2025a). Instead, we use the knowledge of $s$ (which we have already learned), as explored in finite dimensions by Denker et al. (2024).

We can define a denoising score matching loss similar to the finite-dimensional framework. However, we have to be careful that the expressions remain bounded and therefore require an extra assumption.

**Proposition 4.2.** *(Supervised Guidance Training) Define*

$$B := -\frac{e^{-t}}{(1 - e^{-t})^2} \mathbb{E} \left[ \|\mathbb{E}[X_0 \mid X_t, Y] - X_0\|_K^2 \right]. \quad (22)$$

*Then if $B$ is bounded, the minimiser of the score matching objective in (21) is the same as the minimiser of the denoising score matching objective minimising over functions $u$, with expectation taken over $t \sim \mathcal{U}[0, T]$, $X_0, Y \sim \text{Law}(X_0, Y)$, $X_t \sim \text{Law}(X_t \mid X_0)$*

$$\mathbb{E} \left[ \left\| \frac{X_t - e^{-\frac{t}{2}} X_0}{1 - e^{-t}} + s(t, X_t) + u(t, X_t, Y) \right\|_K^2 \right]. \quad (23)$$

*Furthermore, the minimiser $u^*$ is almost surely unique and is given by*

$$u^*(t, x, y) = C\nabla \log h^y(t, x) \quad (24)$$

The proof is similar to Proposition 4 in Baldassari et al. (2023) or Lemma 7 in Pidstrigach et al. (2024) and is provided in Section A.3. We note that $B$ is bounded if one of the two conditions are met:

1. Setting 1 holds, $K$ is equal to the Cameron-Martin space of $C$, and $\mathbb{E}[\|X_0 - \mathbb{E}[X_0]\|_K^2] < \infty$;
2. Setting 2 holds, and the support of $\pi^y$ and $\mathcal{N}(0, C_\pi)$ is contained in $K$.

This is a consequence of Lemma 8 from Pidstrigach et al. (2024), where the proof holds verbatim if we swap the conditional probability $\mathbb{E}[X_0 \mid X_t]$ with $\mathbb{E}[X_0 \mid X_t, Y]$.

We refer to this training framework as supervised guidance training (SGT). The full training setting is provided in Algorithm 1. The SGT loss only requires the forward pass of the unconditional score model. We highlight this in Line 7 of the algorithm, but note that the `stopgrad` is not necessary for the implementation.

*Remark* 4.3. The loss function (23) requires samples $(X_0, Y)$ from the joint distribution. For inverse problems, these pairs are available if the forward operator $G$ and the noise model $\eta$ are known, as one can generate synthetic measurements via $Y = G(X_0) + \eta$. Crucially, this only requires access to the ground truth data $X_0$, which can be sourced either from the original training set or by sampling from the pre-trained score model.

### 4.4. Parametrisation of the Guidance Term

We parametrise the control $u$ using a neural network. In finite-dimensional settings it has been observed that advantageous to make use of a *likelihood-informed inductive bias* and consider parametrisation of the control $u_\phi$ as

$$u_\phi(t,x_t,y) = u_\phi^1(t,x_t,y) + u_\phi^2(t)\nabla_f\Phi(f,y)\big|_{f=\hat{x}_t(x_t)}, \quad (25)$$

where $\nabla_f\Phi(f,y)$ is related to the approximated guidance term in (20). However, crucially, we approximate the Fréchet derivative of the Tweedie map as the identity, drastically reducing the computational time. Here, $u_\phi^1$ is a neural network mapping into $\mathcal{H}$ initialised as zero, and $u_\phi^2 : \mathbb{R} \to \mathbb{R}$ is a time-dependent scalar scaling factor, initialised to a small constant. Similar parametrisations have been successfully applied in finite-dimensional inverse problems (Denker et al., 2024), sampling methods (Phillips et al., 2024; Vargas et al., 2023; Zhang & Chen, 2022), or diffusion model fine-tuning (Venkatraman et al., 2024).

We also encode regularity assumptions directly into the architecture. By preconditioning $u_\phi$ by the covariance operator $C$, we obtain $C\nabla\log h^y(t,x) \approx C^k u_\phi(t,x,y)$, where we study three cases $k \in \{0, 1/2, 1\}$:

- $k = 1$: In this setting, the network $u_\phi$ represents a *pre-score* function. The application of $C$ ensures that the final term $Cu_\phi$ possesses a high regularity,
- $k = 1/2$: In this setting, the network output is preconditioned such that $C^{1/2}u_\phi$ naturally lies in the Cameron-Martin space. In this setting, the network operates in the same space as the driving noise of the SDE,
- $k = 0$: The network is tasked with directly approximating the guidance term and $u_\phi$ must implicitly learn the smoothing properties of the operator $C$.

## 5. Experiments

We evaluate the proposed SGT framework. We compare against other function-space diffusion baselines. The first is the conditional diffusion approach of (Baldassari et al., 2023), in which a task-specific conditional diffusion model is trained from scratch. Under the assumption of optimal training, this serves as an effective upper bound on achievable performance. The second baseline is the FunDPS guidance (Yao et al., 2025), implemented using the approximate guidance formulation in (20).

### 5.1. Implementation Differences in Function Space

In this section we want to show what can be gained from looking at the conditional sampling problem from an infinite-dimensional perspective. That is, we go through some of the implementation differences between the infinite and finite dimension conditioning problems. In particular, there are more choices to consider, including:

1. The choice of discretisation.
2. The choice of the covariance operator $C$.
3. The choice of the Hilbert space to use for the loss.

The first choice is the discretisation of the setup. Looking at the infinite-dimensional problem means we can choose the discretisation that best works for the data we have. This comes down to choosing a basis, or discretising on a grid as would be the usual choice for finite dimensions. Moreover, this choice can be built into the neural network architecture as with neural operators (Li et al., 2019), leading to resolution invariant networks.

The covariance operator $C$ in the conditional case should be chosen in line with the unconditional training. As we see from Theorem 3.1, the conditional SDE $Z_t \mid Y = y$, uses the same covariance operator $C$ as in the unconditional case. Hence, if the unconditional has already been trained, the choice of $C$ is set. For choice of $C$ in the unconditional case, we refer to Section 5 and 6 of Pidstrigach et al. (2024).

Thirdly, we need to choose which $K$ we use in the loss. From Proposition 4.2 we see that in Setting 1 it makes sense to take $K$ to be the Cameron-Martin space corresponding to $C$. In Setting 2, we just need that the support of $\pi$ and $\mathcal{N}(0, C)$ is in $K$, so $H$ is a valid option.

In the experiments, we adopt the following choices. As a base space we consider $L^2(D, \mathbb{R}^d)$, for some choice of domain $D$ and dimension $d$. Given a basis $e_i$ of this space, we consider covariance operators $C_\nu$ of the form

$$C_\nu f := \sum_{k=1}^{\infty} \frac{1}{k^\nu}\langle f, e_k\rangle e_k. \quad (26)$$

### 5.2. Sparse Observations

We consider a one-dimensional inverse problem on $\Omega = [0, 1]$, where the goal is to recover a real-valued function from finitely many noisy point observations. We generate

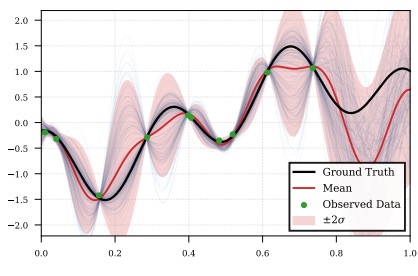 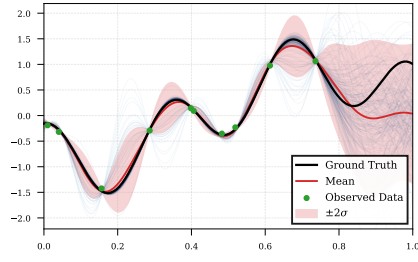 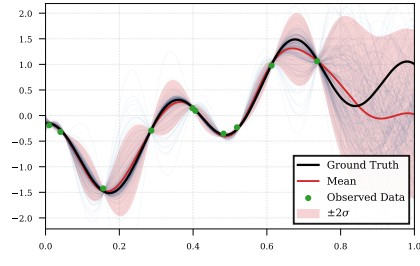

*(a)* Diffusion Posterior Sampling        *(b)* Conditional Diffusion        *(c)* Supervised Guidance Training

*Figure 1.* Posterior sampling for the point evaluation. We show the ground truth signal in black, the mean sample in red and individual samples in light blue. Note, that the uncertainty is small on the observed points (green). SGT is evaluated with $k = 0$ for the preconditioner.

functions as random superpositions of sinusoidal components with varying amplitudes, frequencies, and phases, yielding smooth and bounded functions. Pointwise observations are approximated by integration against a narrow Gaussian kernel, and additive Gaussian noise with standard deviation $\sigma = 0.01$ is applied. Additional details are provided in Appendix C.1.

All score models are based on a Fourier Neural Operator (FNO) architecture (Li et al., 2021), augmented with time embeddings as in Yang et al. (2025a). Notably, SGT uses a smaller network with $8\times$ fewer parameter and is trained for half of the gradient steps compared to the conditional diffusion (full size). We also compare training the conditional diffusion with approximately the same number of parameters and training steps as SGT. We evaluate SGT under three choices of the preconditioner $C^k$ with $k \in \{0, \frac{1}{2}, 1\}$ (see Section 4.4).

To assess the quality of the estimated posterior, we evaluate the approaches on 64 independent signals, drawing $N = 200$ samples per observation. We report both the mean root mean square error (RMSE) and mean energy score (ES) (Gneiting & Raftery, 2007), see Appendix C.1. Quantitative results are given in Table 1 with samples in Figure 1. We observe that SGT is able to beat the FunDPS approximation and comes close to a specifically trained full size conditional model. We also see that when the conditional diffusion is trained with approximately the same number of parameters and training steps as SGT, then SGT beats the conditional model. However, note that full SGT sampling involves both the unconditional model and the guidance model. Hence, the total number of parameters used during inference in SGT (unconditional and guidance networks) is larger than that of the conditional diffusion model. Nonetheless, whereas we only need to re-train the guidance network for a new observation setting, the conditional network has to be trained from scratch. Finally, we see that including the smoothness preconditioner into the architecture is able to produce better results, with $k = 1$ giving the best results for SGT.

**Imperfect Score** We evaluate SGT and FunDPS for three different base models: 1) untrained base model (randomly

*Table 1.* Comparison of the different posterior sampling methods for sparse observations for 64 different signals. (Mean $\pm$ Std)

| Method | RMSE ($\downarrow$) | ES ($\downarrow$) |
|---|---|---|
| FunDPS | $0.456 \pm 0.258$ | $3.709 \pm 2.109$ |
| Cond. Diff. (full size) | $0.347 \pm 0.213$ | $2.766 \pm 1.663$ |
| Cond. Diff. (SGT size) | $0.406 \pm 0.220$ | $3.230 \pm 1.749$ |
| SGT $k = 0$ | $0.390 \pm 0.214$ | $3.148 \pm 1.733$ |
| SGT $k = 1/2$ | $0.385 \pm 0.218$ | $3.111 \pm 1.748$ |
| SGT $k = 1$ | $0.373 \pm 0.210$ | $3.009 \pm 1.645$ |

*Table 2.* Performance under an imperfect base model for FunDPS and SGT.

| | **ES** ($\downarrow$) | **RMSE** ($\downarrow$) |
|---|---|---|
| *Random base model* | | |
| FunDPS ($\gamma = 0.1$) | 20.685 | 558.25 |
| SGT | 6.591 | 1.238 |
| *Partly trained base model* | | |
| FunDPS ($\gamma = 1.0$) | 3.816 | 0.472 |
| SGT | 3.230 | 0.397 |
| *Fully trained base model* | | |
| FunDPS ($\gamma = 1.0$) | 3.608 | 0.447 |
| SGT | 3.190 | 0.392 |

initialised), 2) partly trained base model (50 epochs) and 3) the fully trained model. Quantitative results are presented in Table 2. The performance of both approaches degrades under a worse unconditional base model. However, SGT can partly compensate for a imperfect unconditional model due to the training of the guidance term and is able to obtain a better ES and RMSE for the partly trained model than FunDPS for a fully trained model. Even in the case of a fully random base model SGT is able to roughly recover the true function, whereas FunDPS fails to produce a realistic sample, see Figure 8 in the Appendix.

### 5.3. Heat Equation

Next, we consider a classical Bayesian inverse problem involving the one-dimensional heat equation. The heat equa-

*Table 3.* Comparison of the different posterior sampling methods on the heat equation for 64 different signals. (Mean ± Std)

| Method | RMSE (↓) | ES (↓) |
|---|---|---|
| FunDPS | $0.0519 \pm 0.0311$ | $0.403 \pm 0.213$ |
| Cond. Diff. (full size) | $0.0450 \pm 0.0244$ | $0.351 \pm 0.182$ |
| Cond. Diff. (SGT size) | $0.0483 \pm 0.026$ | $0.383 \pm 0.183$ |
| SGT $k=0$ | $0.0443 \pm 0.0240$ | $0.346 \pm 0.177$ |
| SGT $k=1/2$ | $0.0443 \pm 0.0231$ | $0.345 \pm 0.167$ |
| SGT $k=1$ | $0.0448 \pm 0.0235$ | $0.345 \pm 0.174$ |

*Table 4.* Shape inpainting results for different numbers of EFD coefficients.

| | RMSE (↓) | RMSE obs. (↓) | ES (↓) |
|---|---|---|---|
| **2 coefficients** | | | |
| FunDPS | $2.40 \pm 0.42$ | $0.127 \pm 0.034$ | $1.367 \pm 0.413$ |
| Cond. Diff. | $1.60 \pm 0.44$ | $0.032 \pm 0.005$ | $0.982 \pm 0.036$ |
| SGT (**ours**) | $1.60 \pm 0.39$ | $0.031 \pm 0.004$ | $0.926 \pm 0.295$ |
| **4 coefficients** | | | |
| FunDPS | $1.80 \pm 0.40$ | $0.135 \pm 0.035$ | $0.997 \pm 0.289$ |
| Cond. Diff. | $0.72 \pm 0.09$ | $0.033 \pm 0.004$ | $0.430 \pm 0.075$ |
| SGT (**ours**) | $0.63 \pm 0.09$ | $0.036 \pm 0.004$ | $0.366 \pm 0.081$ |
| **6 coefficients** | | | |
| FunDPS | $0.79 \pm 0.16$ | $0.065 \pm 0.012$ | $0.421 \pm 0.125$ |
| Cond. Diff. | $0.46 \pm 0.06$ | $0.037 \pm 0.004$ | $0.244 \pm 0.054$ |
| SGT (**ours**) | $0.45 \pm 0.06$ | $0.037 \pm 0.004$ | $0.236 \pm 0.050$ |
| **8 coefficients** | | | |
| FunDPS | $0.64 \pm 0.12$ | $0.055 \pm 0.010$ | $0.332 \pm 0.092$ |
| Cond. Diff. | $0.50 \pm 0.05$ | $0.035 \pm 0.004$ | $0.265 \pm 0.036$ |
| SGT (**ours**) | $0.42 \pm 0.04$ | $0.038 \pm 0.003$ | $0.212 \pm 0.035$ |

tion over the domain $\Omega = [0, 1]$ is given by

$$\frac{\partial w}{\partial t} = \nu \frac{\partial^2 w}{\partial x^2}, \quad w(x, 0) = f(x), \tag{27}$$

where $\nu > 0$ is the diffusion coefficients, $f \in L^2(\Omega)$ is the initial temperature and we impose a zero Dirichlet boundary. The inverse problem aims to recover the $f$ from noisy observations at time $T$. We define the forward operator $G : f \mapsto w(\cdot, T)$, such that $y = G(f) + \eta$, where $\eta \sim \mathcal{N}(0, \sigma^2 I)$ represents additive Gaussian noise with $\sigma = 0.1$. We set $\nu = 0.05$ and $T = 0.2$. The training data consists of one-dimensional signals formed by the difference of two symmetric Gaussian bumps with randomly varying location, width, and amplitude.

Quantitative results are summarised in Table 3. We observe that both the conditional diffusion model and our learned guidance function significantly outperform FunDPS in terms of RMSE, with our method achieving competitive accuracy despite a shorter training schedule. However, the effect of the preconditioner $C^k$ is negligible for this operator. Qualitative samples are visualised in Figure 2.

### 5.4. Shape Inpainting

We study conditional sampling of shapes extracted from the MNIST dataset (LeCun et al., 2010). The binary images are converted into closed contours by extracting the outer boundary and sampling points uniform along the curve. We focus on the digit class "3" and its geometric variability. Example shapes from the test set are shown in Figure 5 in Appendix C.3. Each shape is represented by ordered landmarks $(x_i, y_i), i = 1, \ldots, R$ sampled along the contour, where $R$ may vary across examples. Since landmark resolution is a discretisation choice rather than an intrinsic property of the shape, we seek a representation that is independent of $R$. To this end, we embed shapes into a function space by interpreting them as closed parametric curves $(x(t), y(t)), t \in [0, T]$, and represent these curves using elliptic Fourier descriptors (EFDs) (Kuhl & Giardina, 1982), which encode each closed contour as a vector of Fourier coefficients $\alpha = (a_n, b_n, c_n, d_n)_{n=1}^N \in \mathbb{R}^{4N}$, see Appendix C.3. EFD provides a finite-dimensional shape representation that is invariant to landmark resolution.

We formulate conditional sampling as an inverse problem in EFD space. Observations consist of the first $m < N$ Fourier modes, $y = P_m \alpha + \eta$, where $P_m$ projects onto the lowest $m$ modes and $\eta$ denotes additive noise. This corresponds to an inpainting problem in Fourier space; low-frequency shape information is observed, while high-frequency coefficients encoding finer geometric details are missing.

We again compare a conditional diffusion model, the FunDPS approximation and our SGT. Performance is evaluated using the RMSE over the shape, the RMSE over the observed EFD coefficients and the ES. Results are reported in Table 4. Our learned guidance term is perform similar to the conditional diffusion model, in particular outperforming it in the case of only 2 and 4 observed coefficients. FunDPS exhibits higher error and reduced consistency with the observed coefficients. Training and implementation details are provided in Appendix C.3. Qualitative reconstructions are shown in Figure 3. Both the conditional diffusion model and the learned $h$-transform produce realistic and coherent shape completions, whereas FunDPS displays higher variance and struggles to recover fine-scale geometric details. Additional qualitative results are provided in the Appendix.

## 6. Conclusion

In this work, we present a framework for conditional sampling in infinite-dimensional function spaces. In particular, we propose *Supervised Guidance Training*, as an extension of DEFT (Denker et al., 2024) to infinite-dimensional SDMs. We demonstrate that explicitly learning the guidance term leads to a better accuracy, compared to heuristic Tweedie approximations. An assumption of our framework

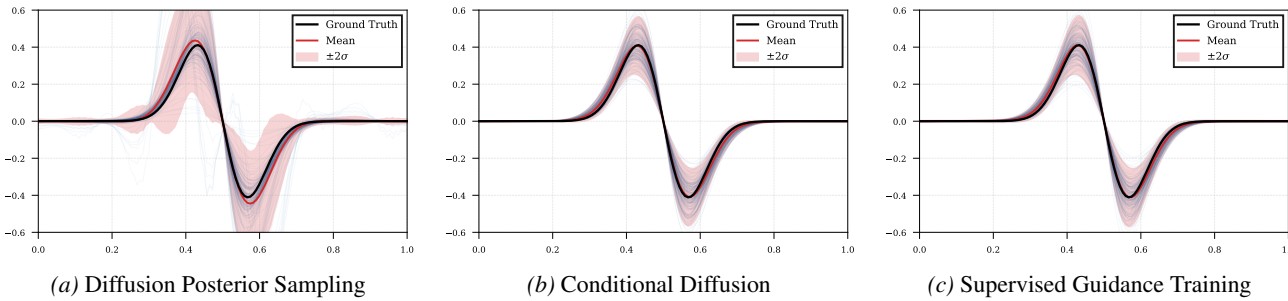

*(a)* Diffusion Posterior Sampling   *(b)* Conditional Diffusion   *(c)* Supervised Guidance Training

*Figure 2.* Posterior sampling for the heat equation. We show the ground truth signal in black, the mean sample in red and individual samples in light blue. Note, that due to the boundary condition $f(0) = f(1) = 0$ the uncertainty at the boundary of the domain is small.

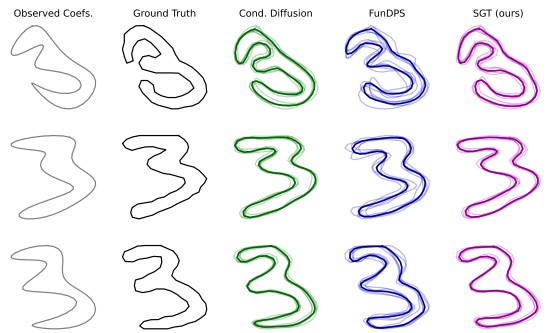

*Figure 3.* Conditional sampling results on shape data using the first six EFD modes. We show the reconstruction from the first six EFD modes, the ground-truth shape, and conditional samples generated by Conditional Diffusion, FunDPS, and our SGT. Mean predictions are shown in darker colours, with individual samples overlaid in lighter shades.

is the availability of an exact score function for the unconditional process, which recovers the prior measure. In practice, score-matching leads to approximation errors in the unconditional score, which can propagate through the guidance term. While the infinite-dimensional framework can naturally deal with discretisation errors, studying how this approximation error influences the final approximation of the posterior remains a subject for further work.

## Acknowledgments

The authors would like to thanks Jakiw Pidstrigach for useful discussions. AD acknowledges support from the EPSRC (EP/V026259/1) and support from DESY (Hamburg, Germany), a member of the Helmholtz Association HGF. EB and JF were supported by funding from Villum Foundation Synergy project number 50091 entitled "Physics-aware machine learning" as well as the Center for Basic Machine Learning Research in Life Science (MLLS) through the Novo Nordisk Foundation (NNF20OC0062606). JF was further supported by funding from the Reinholdt W. Jorck og Hustrus Fond.

## Impact Statement

Our work offers advancements in conditioning diffusion models for functions. This has applications in the field of PDEs, which can be applied to areas with potential societal consequences. Given the theoretical nature of this work, its potential societal impact is indirect and mediated through future applications, making it difficult to anticipate specific consequences at this stage.

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

# A. Proofs

## A.1. Proof of Theorem 3.1

**Theorem A.1.** *Let $\mathbb{P}$ be the path measure of $Z_t$, and $h^y$ defined in (10). The regular conditional probability $\mathbb{P}(\cdot \mid Y = y)$ denoted $\mathbb{P}^y$ satisfies*

$$\mathrm{d}\mathbb{P}^y = h^y(T, Z_T)\mathrm{d}\mathbb{P}. \tag{28}$$

*Moreover, the conditional process of $Z_t$ given $Y = y$, which we denote by $Z_t^y$, satisfies*

$$\mathrm{d}Z_t^y = b(t, Z_t^y)\mathrm{d}t + \sqrt{C}\mathrm{d}W_t;\ Z_0^y \sim \text{Law}(X_T), \tag{29}$$

$$b(t, z) := \frac{1}{2}z + s(T - t, z) + C\nabla \log h^y(t, z), \tag{30}$$

*where $\nabla \log h^y(t, x)$ is the Riesz representation of the Fréchet derivative of the logarithm of $h^y$ with respect to $x$.*

We split the proof of this into a series of lemmas. First we show that we can define the probability measure $\mathbb{P}^y$ via Doob's $h$-transform. For this we require that the function $h^y$ is Fréchet differentiable. We show this is true in Proposition A.2. Next, we prove that this is the "correct" measure, i.e. , $\mathbb{P}^y$ is the conditional measure $\mathbb{P}(\cdot \mid Y = y)$, in Proposition A.5.

**Proposition A.2.** *Let $\mathbb{P}$ be the path distribution of $Z_t$, and $h^y$ as defined in (10). We assume we are either in Setting 1 or Setting 2. Then we can define another distribution $\mathbb{Q}$ via*

$$\mathrm{d}\mathbb{Q} = h^y(T, Z_T)\mathrm{d}\mathbb{P}.$$

*Moreover, writing $Z_t$ with respect to the measure $\mathbb{Q}$ satisfies (29).*

*Proof.* We will apply Theorem 5.1 from Baker et al. (2024), which says that if

1. $h^y$ is twice Fréchet differentiable with respect to $z \in H$ and once differentiable with respect to $t$, with continuous derivatives,

2. the time-reversal SDE in (7) has a strong solution,

3. $h^y(t, Z_t)$ is a strictly positive martingale, with $h^y(0, Z_0) = 1$, and $\mathbb{E}[h^y(T, Z_T)] = 1$,

then $\mathbb{Q}$ is well-defined and $Z_t$ satisfies (29) under $\mathbb{Q}$. We note however that the proof in Theorem 5.1 (Baker et al., 2024) only relies on $h^y$ being once Fréchet differentiable with respect to $z$, therefore this is what we will use.

That the time-reversal has a strong solution holds from Theorem 12 and Theorem 13 of Pidstrigach et al. (2024), for Setting 1 or Setting 2 respectively. Moreover, following Lemma 5.2 of Baker et al. (2024) and noting that the proof does not depend on the SDE having a deterministic initial condition, we know that $h^y(t, Z_t)$ is a strictly positive martingale.

Hence, the main thing to check is that $h^y$ is twice Fréchet differentiable with respect to $z$ and once with respect to time. The differentiability with respect to $z$ holds in Setting 2 from Lemma A.3 and in Setting 1 from Lemma A.4. The time differentiability follows from Lemma 5.3 of Baker et al. (2024). $\square$

**Lemma A.3.** *Let Setting 2 hold. Here, the prior distribution $\pi$ is absolutely continuous with respect to a Gaussian distribution $\mathcal{N}(0, C_\pi)$ such that*

$$\frac{\mathrm{d}\pi}{\mathrm{d}\mathcal{N}(0, C_\pi)}(x) = \frac{\exp(-\Psi(x))}{\mathbb{E}_\pi[\exp(-\Psi(X))]}. \tag{31}$$

*Similarly, we take*

$$\frac{\mathrm{d}\pi^y}{\mathrm{d}\pi}(x) = \frac{\exp(-\Phi(x))}{\mathbb{E}_{\pi^y}[\exp(-\Phi(X))]}. \tag{32}$$

*Moreover, we assume that both $\Phi$ and $\Psi$ are both Fréchet differentiable and that $\mathbb{E}_\pi[\exp(-\Psi(X))] < \infty$, $\mathbb{E}_\pi[\exp(-\Phi(X))] < \infty$.*

*Then $h^y(t, x)$ is Fréchet differentiable with respect to $x$.*

*Proof.* The idea of this proof is to use that the prior distribution $\pi$ has a density with respect to a Gaussian distribution. This is similar to how Pidstrigach et al. (2024, Theorem 13) prove local Lipschitz continuity of the unconditional score. Using similar arguments, we can express everything in terms of the Ornstein-Uhlenbeck SDE started from $\mathcal{N}(0, C_\pi)$, which has a Gaussian form. Let

$$\mathrm{d}V_t = -\frac{1}{2}V_t\mathrm{d}t + \sqrt{C}\mathrm{d}W_t^{\mathcal{H}}, \quad V_0 \sim \mathcal{N}(0, C_\pi), \tag{33}$$

which has the same form as $X_t$, except that $V_0 \sim \mathcal{N}(0, C_\pi)$. Then:

$$V_t = e^{-\frac{t}{2}}V_0 + \sqrt{(1 - e^{-t})}\epsilon, \tag{34}$$

for $\epsilon \sim \mathcal{N}(0, C)$ (N.B. we do not assume $C$ is equal to $C_\pi$). From standard properties of Gaussian variables we therefore know

$$V_0 \mid V_t \sim \mathcal{N}(K_t V_t, P_t), \tag{35}$$

where

$$C_t = e^{-t}C_\pi + (1 - e^{-t})C \qquad K_t = e^{-\frac{t}{2}}C_\pi C_t^{-1}, \qquad P_t = C_\pi - e^{-t}C_\pi C_t^{-1}C_\pi. \tag{36}$$

Furthermore, we know that

$$\mathbb{E}_\pi\left[\exp\left(-\Phi(X_0)\right) \mid X_t = x\right] = \frac{\mathbb{E}\left[\exp\left(-\Psi(V_0)\right)\exp\left(-\Phi(V_0)\right) \mid V_t = x\right]}{\mathbb{E}\left[\exp\left(-\Psi(V_0)\right) \mid V_t = x\right]} \tag{37a}$$

$$= \frac{\int \exp\left(-\Psi(v) - \Phi(v)\right)\mathrm{d}\mathcal{N}(K_t x, P_t)(v)}{\int \exp(-\Psi(v))\mathrm{d}\mathcal{N}(K_t x, P_t)(v)} \tag{37b}$$

$$= \frac{\int \exp\left(-\Psi(v + K_t x) - \Phi(v + K_t x)\right)\mathrm{d}\mathcal{N}(0, P_t)(v)}{\int \exp\left(-\Psi(v + K_t x)\right)\mathrm{d}\mathcal{N}(0, P_t)(v)}. \tag{37c}$$

Therefore, $h^y$ is differentiable with respect to $x$ if $\Phi$ and $\Psi$ are. $\qquad\square$

**Lemma A.4.** *Let the support of $\pi$ lie in the Cameron-Martin space of $C$, such as in Setting 1. Then $h^y(t, x)$ is Fréchet differentiable with respect to $x$.*

*Proof.* The proof for this is similar to the proof that $s(t, x)$ is Lipschitz continuous in Theorem 12 by Pidstrigach et al. (2024). Let $X_t$ be a solution to the forward SDE in (6). Then given $X_0 \in H$, we know that the transition kernel from $X_0$ at time 0 to time $t$ is given by

$$\mathcal{N}(e^{-\frac{t}{2}}X_0, (1 - e^{-t})C). \tag{38}$$

Define

$$n_t(x_0, x_t) := \frac{\mathrm{d}\mathcal{N}(e^{-\frac{t}{2}}x_0, (1 - e^{-t})C)}{\mathrm{d}\mathcal{N}(0, (1 - e^{-t})C)}(x_t). \tag{39}$$

By the Cameron-Martin theorem, which we may apply as we assume the support of $\pi$ lies in the Cameron-Martin space $U$ of $C$, we get

$$n_t(x_0, x_t) = \exp\left(\frac{\langle e^{-\frac{t}{2}}x_0, x_t\rangle_U - e^{-t}\|x_0\|_U^2}{1 - e^{-t}}\right). \tag{40}$$

By the proof of Pidstrigach et al. (2024, Theorem 12) $n_t(x_0, x_t)$ is the joint distribution of $X_0, X_t$.

Moreover, for a function $g$, following the same proof of Pidstrigach et al. (2024, Theorem 12) but replacing $x_0$ with $g(x_0)$ we get

$$\mathbb{E}[g(X_0) \mid X_t = x] = \frac{\int g(x_0) n_t(x_0, x) \mathrm{d}\pi(x_0)}{\int n_t(x_0, x) \mathrm{d}\pi(x_0)}, \tag{41}$$

almost surely. More precisely, by direct computation it holds

$$\mathbb{E}\left[\delta_A \frac{\int g(x_0) n_t(x_0, X_t) \mathrm{d}\pi(x_0)}{\int n_t(x_0, X_t) \mathrm{d}\pi(x_0)}\right] = \int_A \frac{\int_{\mathcal{H}} g(x_0) n_t(x_0, x_t) \mathrm{d}\pi(x_0)}{\int_{\mathcal{H}} n_t(x_0, x_t) \mathrm{d}\pi(x_0)} \mathrm{d}\mathbb{P}_t(x_t) \tag{42a}$$

$$= \int_{\mathcal{H}} \int_A \frac{\int_{\mathcal{H}} g(x_0) n_t(x_0, x_t) \mathrm{d}\pi(x_0)}{\int_{\mathcal{H}} n_t(x_0, x_t) \mathrm{d}\pi(x_0)} n_t(\tilde{x}_0, x_t) \mathrm{d}\mathcal{N}(0, (1 - e^{-t})C)(x_t) \mathrm{d}\pi(\tilde{x}_0) \tag{42b}$$

$$= \int_A \int_{\mathcal{H}} g(x_0) n_t(x_0, x_t) \mathrm{d}\pi(x_0) \frac{\int_{\mathcal{H}} n_t(\tilde{x}_0, x_t) \mathrm{d}\pi(\tilde{x}_0)}{\int_{\mathcal{H}} n_t(x_0, x_t) \mathrm{d}\pi(x_0)} \mathrm{d}\mathcal{N}(0, (1 - e^{-t})C)(x_t) \tag{42c}$$

$$= \int_A \int_{\mathcal{H}} g(x_0) n_t(x_0, x_t) \mathrm{d}\pi(x_0) \mathrm{d}\mathcal{N}(0, (1 - e^{-t})C)(x_t) = \mathbb{E}[\delta_A g(X_0)]. \tag{42d}$$

Hence we know that

$$h^y(t, x) = \mathbb{E}[\exp(-\Phi(X_0)) \mid X_t = x] \tag{43a}$$

$$= \frac{\int \exp(-\Phi(x_0)) n_t(x_0, x) \mathrm{d}\pi(x_0)}{\int n_t(x_0, x) \mathrm{d}\pi(x_0)}. \tag{43b}$$

Then $h^y(t, x)$ is Fréchet differentiable with respect to $x$ if $n_t(x_0, x)$ is, and indeed by the form of $n_t$ coming from the Cameron-Martin theorem, $n_t$ is Fréchet differentiable in $x$ since the inner product is. $\qquad \square$

**Proposition A.5.** *The process $Z_t^y$ is the conditional process $Z_t \mid Y = y$ and thus converges to $Z_T \sim \pi^y$.*

*Proof.* We show by direct calculation that $\mathbb{Q}$ is the regular conditional probability $\mathbb{P}(\cdot \mid Y = y)$. Let $\{\mathcal{F}_s\}$ be the filtration generated by $X_s$. Let $A \in \mathcal{F}_t$. As before, denote $\xi = \mathbb{E}_\pi[\exp(-\Phi(X_0, y))]$. We also denote $\delta_A$ as the Dirac delta function on a set $A$, with $\delta_A(x) = 1, x \in A$, and $\delta_A(x) = 0, x \notin A$. Then

$$\mathbb{Q}^y(A) = \mathbb{E}_{\mathbb{Q}^y}[\delta_A] \tag{44a}$$

$$= \frac{1}{\xi} \mathbb{E}_{\mathbb{P}}[\delta_A \mathbb{E}[\exp(-\Phi(X_0, y) \mid X_t]] \tag{44b}$$

$$= \frac{1}{\xi} \mathbb{E}_{\mathbb{P}}[\mathbb{E}[\delta_A \mid X_t] \exp(-\Phi(X_0, y)] \tag{44c}$$

$$= \frac{1}{\xi} \mathbb{E}_{\mathbb{P}}[\mathbb{P}(A \mid X_0) \exp(-\Phi(X_0, y)] \tag{44d}$$

$$= \frac{1}{\xi} \int \mathbb{P}(A \mid x_0) \exp(-\Phi(x_0, y)) \mathrm{d}\pi(x_0) \tag{44e}$$

$$= \int \mathbb{P}(A \mid x_0) \mathrm{d}\pi^y(x_0) = \mathbb{P}(A \mid Y = y). \tag{44f}$$

$$\square$$

### A.2. Proof of Theorem 3.2

We prove Theorem 3.2, in two lemmas for the two settings. For a proof under Setting 1 see Lemma A.6 and for a proof under Setting 2 see Lemma A.7.

**Lemma A.6.** *Let Setting 1 hold. Then*

$$C \nabla \log h^y(t, x) = s^y(t, x) - s(t, x). \tag{45}$$

*Proof.* We denote by $\pi_t$ the law of $X_t$, and by $\pi_t^y$ the law of the solution to the forward SDE in (6) but started from $X_0 \sim \pi^y$. First we will show that

$$\mathbb{E}[\exp(-\Phi(X_0)) \mid X_t = x] \propto \frac{\mathrm{d}\pi_t^y}{\mathrm{d}\pi_t}, \tag{46}$$

by showing that $\mathbb{E}[\exp(-\Phi(X_0)) \mid X_t = x]$ satisfies

$$\int_A \mathbb{E}[\exp(-\Phi(X_0)) \mid X_t = x]\mathrm{d}\pi_t(x) = \int_A \mathrm{d}\pi_t^y(x). \tag{47}$$

To see this we note that by Lemma A.4, for a function $g$ we have almost surely

$$\mathbb{E}[g(X_0) \mid X_t = x] = \frac{\int g(x_0)n_t(x_0, x)\mathrm{d}\pi(x_0)}{\int n_t(x_0, x)\mathrm{d}\pi(x_0)}, \tag{48}$$

and

$$\mathbb{E}\left[\delta_A \frac{\int g(x_0)n_t(x_0, X_t)\mathrm{d}\pi(x_0)}{\int n_t(x_0, X_t)\mathrm{d}\pi(x_0)}\right] = \int_A \int_{\mathcal{H}} g(x_0)n_t(x_0, x_t)\mathrm{d}\pi(x_0)\mathrm{d}\mathcal{N}(0, (1 - e^{-t})C)(x_t) \tag{49a}$$

$$= \int_{\mathcal{H}} \int_A g(x_0)\mathrm{d}\mathcal{N}(e^{-\frac{t}{2}}x_0, (1 - e^{-t})C)(x_t)\mathrm{d}\pi(x_0). \tag{49b}$$

Hence,

$$\int_A \mathbb{E}[\exp(-\Phi(X_0)) \mid X_t = x]\mathrm{d}\pi_t(x) = \int_{\mathcal{H}} \int_A \exp(-\Phi(x_0))\mathrm{d}\mathcal{N}(e^{-\frac{t}{2}}x_0, (1 - e^{-t})C)(x_t)\mathrm{d}\pi(x_0). \tag{50}$$

Now noting that by Bayes' formula (Stuart, 2010), $\exp(-\Phi(x_0)) \propto \frac{\mathrm{d}\pi^y}{\mathrm{d}\pi}(x_0)$, we see that (50) is proportional to

$$\int_{\mathcal{H}} \int_A \frac{\mathrm{d}\pi^y}{\mathrm{d}\pi}(x_0)\mathrm{d}\mathcal{N}(e^{-\frac{t}{2}}x_0, (1 - e^{-t})C)(x_t)\mathrm{d}\pi(x_0) = \int_{\mathcal{H}} \int_A \mathrm{d}\mathcal{N}(e^{-\frac{t}{2}}x_0, (1 - e^{-t})C)(x_t)\mathrm{d}\pi^y(x_0) = \pi_t^y(A). \tag{51}$$

Now, we know that

$$\frac{\mathrm{d}\pi_t^y}{\mathrm{d}\pi_t}(x) = \frac{\mathrm{d}\pi_t^y}{\mathrm{d}\mathcal{N}(0, (1 - e^{-t})C)}(x) \cdot \left[\frac{\mathrm{d}\pi_t}{\mathrm{d}\mathcal{N}(0, (1 - e^{-t})C)}\right]^{-1}(x). \tag{52}$$

Taking the logarithm, differentiating and applying $C$ therefore gives

$$C\nabla \log \frac{\mathrm{d}\pi_t^y}{\mathrm{d}\pi_t}(x) = C\nabla \log \frac{\mathrm{d}\pi_t^y}{\mathrm{d}\mathcal{N}(0, (1 - e^{-t})C)}(x) - C\nabla \log \frac{\mathrm{d}\pi_t}{\mathrm{d}\mathcal{N}(0, (1 - e^{-t})C)}(x). \tag{53}$$

Hence, to complete the proof we need to show that

$$C\nabla \log \frac{\mathrm{d}\pi_t^y}{\mathrm{d}\mathcal{N}(0, (1 - e^{-t})C)}(x) - C\nabla \log \frac{\mathrm{d}\pi_t}{\mathrm{d}\mathcal{N}(0, (1 - e^{-t})C)}(x) = s^y(t, x) - s(t, x). \tag{54}$$

By definition

$$s(t, x) = -\frac{1}{1 - e^{-t}}\left(x - e^{-\frac{t}{2}}\mathbb{E}[X_0 \mid X_t = x]\right) \tag{55a}$$

$$= -\frac{1}{1 - e^{-t}}\left(x - e^{-\frac{t}{2}}\frac{\int x_0 n_t(x_0, x)\mathrm{d}\pi(x_0)}{\int n_t(x_0, x)\mathrm{d}\pi(x_0)}\right), \tag{55b}$$

and similarly for $s^y(t, x)$ and so

$$s^y(t, x) - s(t, x) = \frac{e^{-\frac{t}{2}}}{1 - e^{-t}}\left(\frac{\int x_0 n_t(x_0, x)\mathrm{d}\pi^y(x_0)}{\int n_t(x_0, x)\mathrm{d}\pi^y(x_0)} - \frac{\int x_0 n_t(x_0, x)\mathrm{d}\pi(x_0)}{\int n_t(x_0, x)\mathrm{d}\pi(x_0)}\right). \tag{56}$$

Now

$$C\nabla \log \frac{\mathrm{d}\pi_t}{\mathrm{d}\mathcal{N}(0,(1-e^{-t})C)}(x) = C\nabla \log \int n_t(x_0,x)\mathrm{d}\pi(x_0) \tag{57a}$$

$$= \frac{\int C\nabla n_t(x_0,x)\mathrm{d}\pi(x_0)}{\int n_t(x_0,x)\mathrm{d}\pi(x_0)}. \tag{57b}$$

By definition,

$$n_t(x_0,x_t) = \exp\left(\frac{\langle e^{-\frac{t}{2}}x_0, x_t\rangle_U - e^{-t}\|x_0\|_U^2}{1-e^{-t}}\right), \tag{58}$$

so

$$\nabla_x n_t(x_0,x) = \frac{e^{-\frac{t}{2}}}{1-e^{-t}}C^{-1}x_0 n_t(x_0,x). \tag{59}$$

Substituting this back into (57b) gives

$$\frac{e^{-\frac{t}{2}}}{1-e^{-t}}\frac{\int x_0 n_t(x_0,x)\mathrm{d}\pi(x_0)}{\int n_t(x_0,x)\mathrm{d}\pi(x_0)}. \tag{60}$$

Following the same logic for $C\nabla \log \frac{\mathrm{d}\pi_t^y}{\mathrm{d}\mathcal{N}(0,(1-e^{-t})C)}(x)$ and comparing to (56) gives the result. $\qquad\square$

**Lemma A.7.** *Let Setting 2 hold so that $\frac{\mathrm{d}\pi}{\mathrm{d}\mathcal{N}(0,C_\pi)}(x) = \exp(-\Psi(x))$. Then*

$$C\nabla \log h^y(t,x) = s^y(t,x) - s(t,x). \tag{61}$$

*Proof.* Throughout the proof, we simplify notation by writing $\Phi(\cdot)$ instead of $\Phi(\cdot,y)$. Let $V_0, V_t, C_t, K_t, P_t$ be as defined in Lemma A.3. Then from there, we know that

$$h(t,x) = \mathbb{E}_\pi\left[\exp\left(-\Phi(X_0)\right) \mid X_t = x\right] = \frac{\int \exp\left(-\Psi(v+K_t x) - \Phi(v+K_t x)\right)\mathrm{d}\mathcal{N}(0,P_t)(v)}{\int \exp\left(-\Psi(v+K_t x)\right)\mathrm{d}\mathcal{N}(0,P_t)(v)}. \tag{62}$$

To simplify notation let's say that $h(t,x) = \frac{f(x)}{g(x)}$, with

$$f(x) := \int \exp\left(-\Psi(v+K_t x) - \Phi(v+K_t x)\right)\mathrm{d}\mathcal{N}(0,P_t)(v) \tag{63a}$$

$$g(x) := \int \exp\left(-\Psi(v+K_t x)\right)\mathrm{d}\mathcal{N}(0,P_t)(v). \tag{63b}$$

Then

$$Df(x)[a] = -\int \exp\left(-(\Psi+\Phi)(v+K_t x)\right)(D\Psi+D\Phi)(v+K_t x)[K_t a]\mathrm{d}\mathcal{N}(0,P_t)(v) \tag{64a}$$

$$= -\int \exp\left(-(\Psi+\Phi)(v)\right)(D\Psi+D\Phi)(v)[K_t a]\mathrm{d}\mathcal{N}(K_t x, P_t)(v). \tag{64b}$$

Furthermore, we see that

$$\frac{Df(x)[a]}{f(x)} = -\mathbb{E}_{\pi^y}[(D\Psi+D\Phi)(X_0)[K_t a] \mid X_t = x]. \tag{64c}$$

Similarly for $g$

$$Dg(x)[a] = -\int \exp\left(-\Psi(v)\right)D\Psi(v)[K_t a]\mathrm{d}\mathcal{N}(K_t x, P_t)(v), \tag{65a}$$

$$\frac{Dg(x)[a]}{g(x)} = -\mathbb{E}_\pi[D(\Psi)(X_0)[K_t a] \mid X_t = x]. \tag{65b}$$

Now

$$D_x\left[\log h(t,x)\right][a] = D\log\frac{f(x)}{g(x)} = \frac{Df(x)}{f(x)} - \frac{Dg(x)}{g(x)} \tag{66a}$$

$$= \mathbb{E}_\pi[D\Psi(X_0)[K_t a] \mid X_t = x] - \mathbb{E}_\pi[(D\Psi + D\Phi)(X_0)[K_t a] \mid X_t = x]. \tag{66b}$$

By the Riesz representation theorem, let $\nabla\Psi(X_0) \in \mathcal{H}$ be the unique element such that $D\Psi(X_0)[a] = \langle\nabla\Psi(X_0), a\rangle_{\mathcal{H}}$, and similarly for $\nabla\Phi(X_0)$. Then we note that

$$D_x\left[\log h(t,x)\right][a] = \mathbb{E}_\pi[\langle\nabla\Psi(X_0), K_t a\rangle \mid X_t = x] - \mathbb{E}_\pi[\langle(\nabla\Psi + \nabla\Phi)(X_0), K_t a\rangle \mid X_t = x] \tag{67a}$$

$$= \langle\mathbb{E}[\nabla\Phi(X_0) \mid X_t = x], K_t a\rangle. \tag{67b}$$

Hence, $C\nabla\log h(t,x) = CK_t^*\mathbb{E}[\nabla\Phi(X_0) \mid X_t = x]$, where $K_t^*$ is the adjoint of $K_t$.

Now we show that $s^y(t,x) - s(t,x)$ can also be expressed in this way.

By definition,

$$s^y(t,x) - s(t,x) = \frac{e^{-t/2}}{1-e^{-t}}(\mathbb{E}[X_0 \mid X_t = x] - \mathbb{E}[X_0 \mid X_t = x, Y = y]) \tag{68a}$$

$$= \frac{e^{-t/2}}{1-e^{-t}}\left(\frac{\mathbb{E}[V_0\exp(-\Psi(V_0)) \mid V_t = x]}{\mathbb{E}[\exp(-\Psi(V_0)) \mid V_t = x]} - \frac{\mathbb{E}[V_0\exp(-\Phi(V_0) - \Psi(V_0)) \mid V_t = x]}{\mathbb{E}[\exp(-\Phi(V_0) - \Psi(V_0)) \mid V_t = x]}\right). \tag{68b}$$

By Stein's Lemma, for a function differentiable function $G : \mathcal{H} \to \mathbb{R}$ (which will represent combinations of $\Phi, \Psi$) we know

$$\mathbb{E}[V_0\exp(-G(V_0)) \mid V_t = x] = K_t x\,\mathbb{E}[\exp(-G(V_0)) \mid V_t = x] - P_t\mathbb{E}[\exp(-G(V_0))\nabla G(V_0) \mid V_t = x]. \tag{69}$$

Dividing everything by $\mathbb{E}[\exp(-G(V_0)) \mid V_t = x]$ gives

$$\frac{\mathbb{E}[V_0\exp(-G(V_0)) \mid V_t = x]}{\mathbb{E}[\exp(-G(V_0)) \mid V_t = x]} = K_t x - P_t\frac{\mathbb{E}[\exp(-G(V_0))\nabla G(V_0) \mid V_t = x]}{\mathbb{E}[\exp(-G(V_0)) \mid V_t = x]}. \tag{70}$$

Applying this with $G = \Psi$ and $G = \Psi + \Phi$ we therefore see that

$$\mathbb{E}[X_0 \mid X_t = x] = K_t x - P_t\mathbb{E}[\nabla\Psi(X_0) \mid X_t = x], \tag{71a}$$

$$\mathbb{E}[X_0 \mid X_t = x, Y = y] = K_t x - P_t\mathbb{E}[(\nabla\Phi + \nabla\Psi)(X_0) \mid X_t = x], \tag{71b}$$

hence

$$s^y(t,x) - s(t,x) = \frac{e^{-t/2}}{1-e^{-t}}P_t\mathbb{E}[\nabla\Phi(X_0) \mid X_t = x]. \tag{72}$$

Then all that is left to show is that $\frac{e^{-t/2}}{1-e^{-t}}P_t = CK_t^*$. For ease, we repeat the definitions of the operators here:

$$C_t = e^{-t}C_\pi + (1-e^{-t})C, \quad K_t = e^{-\frac{t}{2}}C_\pi C_t^{-1}, \quad P_t = C_\pi - e^{-t}C_\pi C_t^{-1}C_\pi. \tag{73}$$

From this, and using that $C_t^{-1}, C_\pi$ are self-adjoint we see that

$$C = (1-e^{-t})^{-1}(C_t - e^{-t}C_\pi), \tag{74a}$$

$$CK_t^* = \frac{e^{t/2}}{1-e^{-t}}(C_t - e^{-t}C_\pi)C_t^{-1}C_\pi \tag{74b}$$

$$= \frac{e^{t/2}}{1-e^{-t}}(C_\pi - e^{-t}C_\pi C_t^{-1}C_\pi) = \frac{e^{t/2}}{1-e^{-t}}P_t. \tag{74c}$$

$\square$

### A.3. Proof of Proposition 4.2

*Proof.* We can follow the proof of Proposition 4 in Baldassari et al. (2023) for the first part of the proof. It suffices to show the claim for a single $t$, due to the tower rule of expectations. First, we can decompose the score-matching loss function as

$$\mathbb{E}_{X_t,Y}\left[\|s^Y(t,X_t) - [s(t,X_t) + u(t,X_t,Y)]\|^2\right] \tag{75a}$$

$$= \mathbb{E}_{X_t,Y}[\|s^Y(t,X_t)\|^2] + \mathbb{E}_{X_t,Y}[\|s(t,X_t) + u(t,X_t,Y)\|^2] - 2\mathbb{E}_{X_t,Y}[\langle s^Y(t,X_t), s(t,X_t) + u(t,X_t,Y)\rangle]. \tag{75b}$$

Using Lemma 8 in Pidstrigach et al. (2024), we know that $\mathbb{E}_{X_t,Y}[\|s^Y(t,X_t)\|^2]$ is bounded. Further, we can decompose the inner product, using the definition of the conditional score as

$$\mathbb{E}_{X_t,Y}[\langle s^Y(t,X_t), s(t,X_t) + u(t,X_t,Y)\rangle] \tag{76a}$$

$$= -\frac{1}{(1-e^{-t})}\mathbb{E}_{X_t,Y}\left[\left\langle \mathbb{E}[X_t - e^{-\frac{t}{2}}X_0|Y,X_t], s(t,X_t) + u(t,X_t,Y)\right\rangle\right] \tag{76b}$$

$$= -\frac{1}{(1-e^{-t})}\mathbb{E}_{X_t,Y}\left[\mathbb{E}_{X_0}\left[\left\langle X_t - e^{-\frac{t}{2}}X_0, s(t,X_t) + u(t,X_t,Y)\right\rangle \mid X_t,Y\right]\right] \tag{76c}$$

$$= -\frac{1}{(1-e^{-t})}\mathbb{E}_{X_0,X_t,Y}\left[\left\langle X_t - e^{-\frac{t}{2}}X_0, s(t,X_t) + u(t,X_t,Y)\right\rangle\right]. \tag{76d}$$

By completing the square in (75b) we get

$$\mathbb{E}_{X_t,Y}\left[\|s^Y(t,X_t) - [s(t,X_t) + u(t,X_t,Y)]\|^2\right] \tag{77}$$

$$= B + \mathbb{E}_{X_0,X_t,Y}\left[\| -(1-e^{-t})^{-1}(X_t - e^{-\frac{t}{2}}X_0) - [s(t,X_t) + u(t,X_t,Y)]\|^2\right] \tag{78}$$

with

$$B = \mathbb{E}_{X_t,Y}[\|s^Y(t,X_t)\|^2] - \mathbb{E}_{X_0,X_t}\left[\|(1-e^{-t})^{-1}(X_t - e^{-\frac{t}{2}}X_0)\|^2\right], \tag{79}$$

which is independent of $u$. In order for the denoising score matching loss to be finite, we therefore need that the term $B$ is finite.

Following the proof of Lemma 7 in (Pidstrigach et al., 2024), we may further simplify the term $B$. By the tower property of conditional expectations it holds

$$\mathbb{E}[\langle s^Y(t,X_t), (1-e^{-t})^{-1}(X_t - e^{-\frac{t}{2}}X_0)\rangle] \tag{80a}$$

$$= -(1-e^{-t})^{-2}\mathbb{E}[\langle \mathbb{E}[(X_t - e^{-\frac{t}{2}}X_0) \mid X_t,Y], (X_t - e^{-\frac{t}{2}}X_0)\rangle] \tag{80b}$$

$$= -(1-e^{-t})^{-2}\mathbb{E}[\|\mathbb{E}[(X_t - e^{-\frac{t}{2}}X_0) \mid X_t,Y]\|^2] \tag{80c}$$

$$= -\mathbb{E}[\|s^Y(t,X_t)\|^2]. \tag{80d}$$

Then

$$B = -\mathbb{E}[\|s^Y(t,X_t)\|^2] + 2\mathbb{E}\left[\left\langle s^Y(t,X_t), \frac{1}{1-e^{-t}}(X_t - e^{-\frac{t}{2}}X_0)\right\rangle\right] - \mathbb{E}_{X_0,X_t}\left[\left\|\frac{1}{1-e^{-t}}(X_t - e^{-\frac{t}{2}}X_0)\right\|^2\right] \tag{81a}$$

$$= -\mathbb{E}[\|s^Y(t,X_t) - (1-e^{-t})^{-1}(X_t - e^{-\frac{t}{2}}X_0)\|^2] \tag{81b}$$

$$= -\frac{e^{-t}}{(1-e^{-t})^2}\mathbb{E}\left[\|\mathbb{E}[X_0 \mid X_t,Y] - X_0\|_K^2\right]. \tag{81c}$$

Now, if $B$ is bounded, then the denoising score matching loss is, so the minimisers of the explicit score matching and the denoising score matching loss are identical.

Further, as the explicit score matching objective is the expectation of a convex function, i.e. the squared $L^2$ norm, the minimum is achieved only if the integrand is zero almost everywhere. Therefore we get

$$u^*(t,X_t,Y) = s^Y(t,X_t) - s(t,X_t). \tag{82}$$

As the set of measurable $L^2$ function is convex and closed, the minimiser is almost surely unique. Further, we know that $u^*$ is bounded due to assumptions on $B$. $\qquad\square$

# B. Stochastic Optimal Control

In this section, we give some background on the SOC viewpoint discussed in Section 4.1. In particular, for the finite-dimensional this has recently seen use, see e.g. , for model fine-tuning (Domingo-Enrich et al., 2025; Domingo i Enrich et al., 2024), sampling from unnormalised densities (Zhang & Chen, 2022), conditional sampling (Pidstrigach et al., 2025) or inverse problems (Denker et al., 2024).

Let $\mathbb{P}$ be the path measure of the unconditional reverse SDE (7), which we re-state here

$$\mathrm{d}Z_t = \left[\frac{1}{2}Z_t + s(T - t, Z_t)\right]\mathrm{d}t + \sqrt{C}\mathrm{d}W_t^{\mathcal{H}}.$$

Let us define

$$\mathcal{E}(\mathbb{Q}) := \mathrm{KL}(\mathbb{Q}\|\mathbb{P}) + \mathbb{E}_{\mathbb{Q}}[\Phi(Z_T, y)], \tag{83}$$

for all $\mathbb{Q} \ll \mathbb{P}$. Then, we have the following proposition.

**Proposition B.1.** *The conditional path measure $\mathbb{P}^y$ is the unique minimiser of $\mathcal{F}(\mathbb{Q})$.*

*Proof.* We start by rewriting $\mathcal{E}$ as

$$\mathcal{E}(\mathbb{Q}) = \mathbb{E}_{\mathbb{Q}}\left[\log\frac{\mathrm{d}\mathbb{Q}}{\mathrm{d}\mathbb{P}} + \Phi(Z_T, y)\right]. \tag{84}$$

From the representation of $\mathbb{P}^y$ in Theorem 3.1 we have that

$$\Phi(Z_T, y) = -\log\frac{\mathrm{d}\mathbb{P}^y}{\mathrm{d}\mathbb{P}} - \log\xi, \tag{85}$$

with $\xi = \mathbb{E}_{\mathbb{P}}[\exp(-\Phi(Z_T, y)]$ as the normalisation constant. Substituting this in (84) gives us

$$\mathcal{E}(\mathbb{Q}) = \mathbb{E}_{\mathbb{Q}}\left[\log\frac{\mathrm{d}\mathbb{Q}}{\mathrm{d}\mathbb{P}} - \log\frac{\mathrm{d}\mathbb{P}^y}{\mathrm{d}\mathbb{P}} - \log\xi\right] = \mathbb{E}_{\mathbb{Q}}\left[\frac{\mathrm{d}\mathbb{Q}}{\mathrm{d}\mathbb{P}^y}\right] - \log\xi = \mathrm{KL}(\mathbb{Q}\|\mathbb{P}^y) - \log\xi. \tag{86}$$

As the KL divergence is non-negative and only zero if and only if $\mathbb{Q} = \mathbb{P}^y$ we have our result. $\square$

Now, we consider path measures $\mathbb{Q}^u$ that are induced by the reverse SDE

$$dZ_t^u = (b(t, Z_t^u) + Cu(t, Z_t^u))dt + \sqrt{C}dW_t^{\mathbb{Q}^u}, \tag{87}$$

with $b(t, z) = \frac{1}{2}z + s(T - t, z)$ as the drift of (7) and the objective function

$$\mathcal{J}(u) := \mathcal{E}(\mathbb{Q}^u) = \mathrm{KL}(\mathbb{Q}^u\|\mathbb{P}) + \mathbb{E}_{\mathbb{Q}^u}[\Phi(Z_T^u, y)]. \tag{88}$$

Using Girsanov's theorem (Da Prato & Zabczyk, 2014, Theorem 10.14), we can write the Radon-Nikodym derivative as

$$\frac{\mathrm{d}\mathbb{Q}^u}{\mathrm{d}\mathbb{P}} = \exp\left(\int_0^T \langle\sqrt{C}u_t, dW_t^{\mathbb{P}}\rangle - \frac{1}{2}\int_0^T \|\sqrt{C}u_t\|_{\mathcal{H}}^2\mathrm{d}t\right) \tag{89a}$$

$$= \exp\left(\int_0^T \langle\sqrt{C}u_t, \sqrt{C}u_t\rangle\mathrm{d}t + \int_0^T \langle\sqrt{C}u_t, dW_t^{\mathbb{Q}^u}\rangle - \frac{1}{2}\int_0^T \|\sqrt{C}u_t\|_{\mathcal{H}}^2\mathrm{d}t\right) \tag{89b}$$

$$= \exp\left(\int_0^T \langle\sqrt{C}u_t, dW_t^{\mathbb{Q}^u}\rangle + \frac{1}{2}\int_0^T \|\sqrt{C}u_t\|_{\mathcal{H}}^2\mathrm{d}t\right) \tag{89c}$$

as $dW_t^{\mathbb{P}} = \sqrt{C}u_t\mathrm{d}t + dW_t^{\mathbb{Q}^u}$. Substituting this in $\mathcal{J}(u)$ and noting that the Itô integral has expectation zero, we get the stochastic optimal control loss function

$$\mathcal{J}(u) = \mathbb{E}_{\mathbb{Q}^u}\left[\Phi(Z_T^u, y) + \frac{1}{2}\int_0^T \|\sqrt{C}u_t\|_{\mathcal{H}}^2 dt\right]. \tag{90}$$

# C. Experimental Details

The code publicly available at: https://github.com/alexdenker/FuncSGT

## C.1. Sparse Observations

We adapt the architecture of the Fourier Neural Operator (FNO) (Li et al., 2021) to design network architectures which can be applied to any discretisation. The original FNO architecture does not incorporate additional time information, while in our case the score depends explicitly on the time. For this, we make use of a learnable time-embedding, which was previously used for diffusion bridges (Yang et al., 2025a) and PDE applications (Park et al., 2023). The time-independent FNO layer is defined as

$$\mathcal{L}_l(v_t)(x) := \sigma(W_l v(x) + \mathcal{F}^{-1}[R_l \cdot \mathcal{F}(v)](x)), \tag{91}$$

with $\sigma : \mathbb{R} \to \mathbb{R}$ a (non-linear) activation function, $W_l$ a weight matrix and $\mathcal{F}, \mathcal{F}^{-1}$ are the Fourier transform and inverse Fourier transform, respectively, and $R_l$ is a learned filter in Fourier space. Here, both $W_l$ and $R_l$ are learnable parameters of the layer. For all of our experiments we use the sigmoid linear unit as the activation function (Ramachandran et al., 2017). The time-dependent FNO layer, as proposed by Park et al. (2023), is then implemented using two time-modulations $\psi_l(t)$ and $\varphi_l(t)$ as

$$\mathcal{L}_l(v_t)(x, t) := \sigma(W_l \psi_l(t) v(x) + \mathcal{F}^{-1}[\varphi_l(t) R_l \cdot \mathcal{F}(v)](x)). \tag{92}$$

Motivated by the success of adaptive normalisation in diffusion models (Dhariwal & Nichol, 2021), we slightly deviate from this implementation and change the time-modulation in physical space to

$$\mathcal{L}_l(v_t)(x, t) := \sigma((1 + \gamma_l(t))W_l v(x) + \beta_l(t) + \mathcal{F}^{-1}[\varphi_l(t) R_l \cdot \mathcal{F}(v)](x)), \tag{93}$$

and learn both a scaling $\gamma_l(t)$ and a bias $\beta_l(t)$.

**Dataset** We construct a synthetic prior over one-dimensional functions on the interval $[0, 1]$. Each sample is generated as a random superposition of sinusoidal components with random frequencies, amplitudes and phases. In particular, for each sample we draw an integer $n_{\text{terms}} \sim \mathcal{U}\{1, \ldots, 8\}$ and generate

$$f(x) = \sum_{i=1}^{n_{\text{terms}}} a_i \sin(\omega_i \pi x + \phi_i), \tag{94}$$

with amplitudes $a_i \sim \mathcal{U}[0.3, 1.0]$, frequencies $\omega_i \sim \mathcal{U}[1.0, 11.0]$ and phase $\phi_i \sim \mathcal{U}(0, 2\pi)$. This always yields smooth and bounded functions. We generate $10\,000$ independent samples for training. We use $\nu = 1$ for the covariance operator in Equation (26).

**Forward Operator** To generate observations, we define a forward operator that approximated pointwise function evaluation using integration against a narrow Gaussian kernel. Given a function $f(x)$ and a set of locations $\{x_i\}_{i=1}^k$, each point evaluation is computed as

$$G(f, x_i) = \int_0^1 f(x) w_\sigma(x - x_i) dx, \tag{95}$$

$$w_\sigma(z) = \frac{1}{Z} \exp\left(-\frac{z^2}{2\sigma^2}\right), \tag{96}$$

with bandwidth $\sigma = 0.02$. The observations are then given as $y = \{G(f, x_i)\}$ for locations $x_i, i = 1, \ldots, R$. We add i.i.d. Gaussian noise to each point evaluation with a standard deviation $\sigma_\eta = 0.01$. We use $R = 10$ observations.

**Conditional Diffusion Model** The conditional diffusion model depends explicitly on the observations $y$. However, naively concatenating the noisy signal $f_t$ and the observations $y$ does not work, as $f_t$ is discretised on a spatial grid of $n_x$ points

and $y$ are sparse observations on $k$ locations. We therefore lift the sparse measurements onto the same spatial grid via a Gaussian kernel $\kappa$ (bandwidth $\sigma_x = 0.01$). In particular, we construct two additional input channels

$$m(x) = \sum_{i=1}^{k} \kappa(x, x^i), \quad v(x) = \sum_{i=1}^{k} y^i \kappa(x, x^i), \tag{97}$$

where $m(x)$ encodes observation locations and $v(x)$ encodes observed values. The network input is the channel-wise concatenation $[f_t(x), v(x), m(x)]$. This approach allows the conditional diffusion model to generalise across varying sensor numbers and locations.

**Training Setting**    For all methods, we use the ADAM optimiser (Kingma & Ba, 2015) with a cosine learning rate decay starting at $1 \times 10^{-4}$ and ending in $1 \times 10^{-5}$ and a batch size of 256. The unconditional diffusion model is trained for $40\,000$ gradient steps. We use a time-modulated FNO with 4 layers, 16 modes and 64 hidden channels, which results in $1\,140\,161$ trainable parameters. For the conditional diffusion model, we use 8 FNO layers with 16 modes and a hidden channel dimension of 128, which results in $8\,737\,153$ trainable parameters. We train it for $80\,000$ gradient steps. The guidance network is implemented similarly to the unconditional model with 4 layers, 16 modes and 64 hidden channels and it also trained for $40\,000$ gradient steps.

**Evaluation Metrics**    Assume that $f_{\text{gt}}$ is the ground truth signal and we have $N$ samples $\{f^i\}_{i=1}^N$. We compute both the root mean squared error (RMSE)

$$\text{RMSE}(\{f^i\}, f_{\text{gt}}) = \left( \frac{1}{N} \sum_{i=1}^{N} \|f^i - f_{\text{gt}}\|^2 \right)^{1/2} \tag{98}$$

and the energy score (ES)

$$\text{ES}(\{f^i\}, f_{\text{gt}}) = \frac{1}{N} \sum_{i=1}^{N} \|f^i - f_{\text{gt}}\|^\beta - \frac{1}{2N^2} \sum_{i=1}^{N} \sum_{j=1}^{N} \|f^i - f^j\|^\beta \tag{99}$$

with $\beta = 1.0$ (Gneiting & Raftery, 2007). Here, the RMSE measures how close the samples are on average to the ground truth signal and the ES scores how well the distribution is captured.

### C.2. Heat Equation

**Training setting**    For all three training setups, we use the ADAM optimiser (Kingma & Ba, 2015) with a cosine learning rate decay starting at $1 \times 10^{-4}$ and ending in $1 \times 10^{-5}$ and a batch size of 256. The unconditional diffusion model is trained for $10^5$ gradient steps. We use 4 time-modulated FNO layers with 16 modes in the Fourier transform and a channel dimension of 64. In total, the diffusion model has $1\,140\,161$ trainable parameters. We train the conditional diffusion model using the loss function in Baldassari et al. (2023). The conditional diffusion models depends explicitly on the observations $y$. For this experiment both the observation and the ground truth signal are functions over $\Omega$. We discretise both on the same spatial grid and input both as two channels into the FNO. As the conditional diffusion model has to amortise over different observations $y$, we use a larger architecture. In particular, we use 8 layers instead of 4 and the conditional diffusion model has $2\,271\,745$ trainable parameters. We also train the conditional diffusion model for more $2 \times 10^5$ gradient steps, double the amount of the unconditional diffusion model. For the FunDPS (Yao et al., 2025) we have to choose the scaling parameter, i.e. , $\gamma > 0$ in (20). We choose this via a grid search to minimise the RMSE, the final value was $\gamma = 1.3$. Our learned guidance term is implemented using the parametrisation from (25). Here, $u_\phi^1$ is implemented using the same architecture as the unconditional diffusion model and $u_\phi^2$ is a simple two layer neural networks. The guidance model is trained for $2 \times 10^4$ gradient steps.

**Training Data**    We discretise the domain $\Omega$ with 128 equidistant points. The ground truth signals are sampled from a parametrised distribution

$$f(x) = \alpha \left( \exp\left(-\gamma(x-\lambda)^2\right) - \exp\left(-\gamma(x-(1-\lambda))^2\right) \right),$$

with $\lambda \sim \mathcal{U}[0.2, 0.5]$, $\gamma \sim \mathcal{U}[75, 115]$ and $\alpha \sim \mathcal{U}[0.9, 1.1]$. This corresponds to a distribution of simple functions with two Gaussian bumps. We solve the heat equation via the Crank-Nicolson method with $dt = 0.001$ (Crank & Nicolson, 1947) to simulate measurements. We use $\nu = 1$ for the covariance operator in Equation (26).

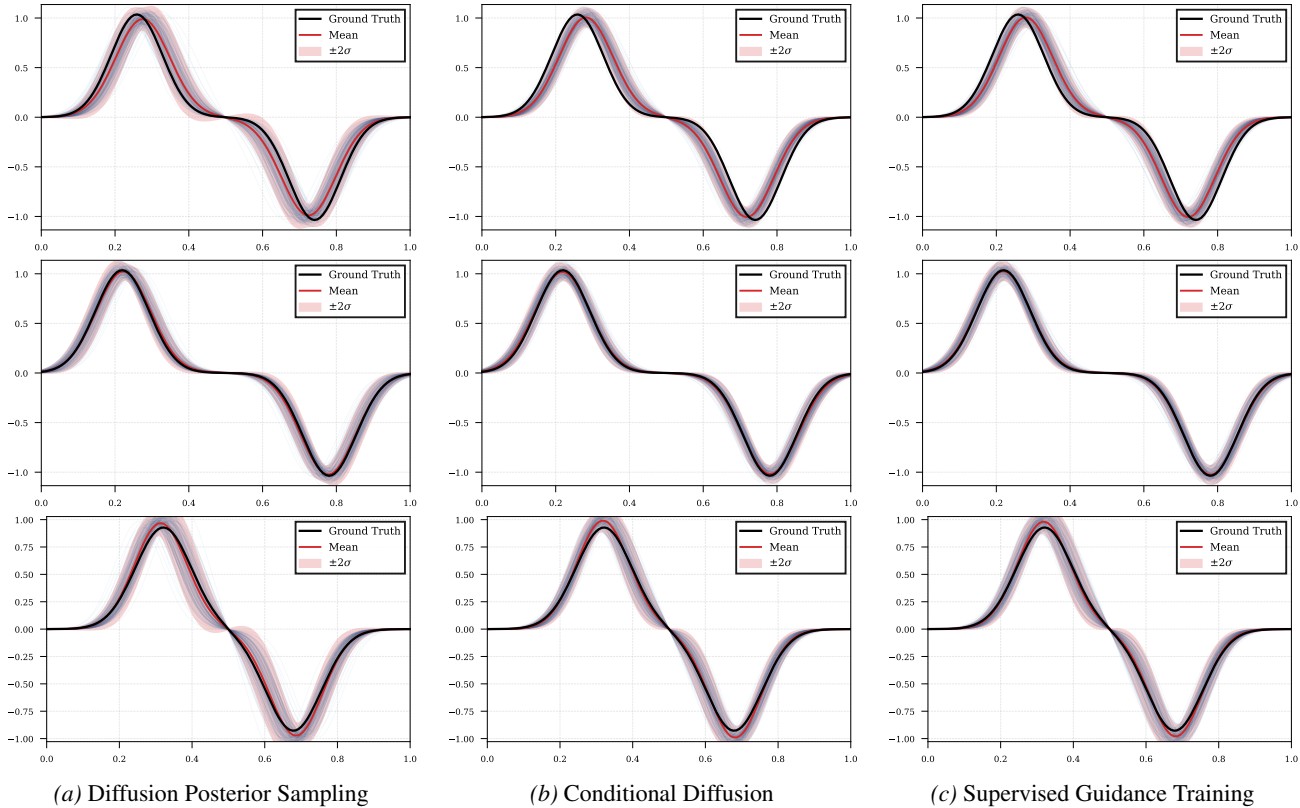

*(a)* Diffusion Posterior Sampling  *(b)* Conditional Diffusion  *(c)* Supervised Guidance Training

*Figure 4.* Posterior sampling for the heat equation. We show the ground truth signal in black, the mean sample in red and individual samples in light blue. Note, that due to the boundary condition $f(0) = f(1) = 0$ the uncertainty at the boundary of the domain is small.

### C.3. Shape Inpainting

**Elliptic Fourier descriptors**  We make use of elliptic Fourier descriptors (EFD) (Kuhl & Giardina, 1982). The curve $(x(t), y(t)), t \in [0, T]$ is expanded as

$$x(t) = a_0 + \sum_{n=1}^{N} (a_n \cos(nt) + b_n \sin(nt)), \tag{100}$$

$$y(t) = c_0 + \sum_{n=1}^{N} (c_n \cos(nt) + d_n \sin(nt)), \tag{101}$$

where centering removed the zeroth-order terms. This results in a finite-dimensional representation of the curve, suitable as an input to a neural network.

**Shape Network**  We implement the diffusion models using a similar design principle as the FNO (Li et al., 2021). Given a noisy shape represented by landmarks, the model acts in three stages. First, we convert the noisy landmarks into the EFD coefficients, where we use $N = 16$ modes. The score function $s_\theta(\cdot, t)$ is then parametrised directly in Fourier space by a small residual multi-layer perceptron (MLP). The time $t$ is incorporated as an additional input channel via a time embedding. The predicted score in Fourier space is mapped back to the landmark space using an inverse Fourier transform, producing a vector field on the landmark coordinates. This architecture ensures that all learned operations act on fixed-dimensional coefficients, while making the model compatible with variable-resolution landmarks input. We train the unconditional network using the ADAM optimiser, using a fixed learning rate of $1 \times 10^{-3}$ for 5000 epochs. We train the conditional diffusion model using the same architecture as the unconditional model with the conditional information, i.e. , the first 5 EFD modes, appended to the input, which results in $s_\theta([x_t, c], t)$. We discretise the reverse SDE using Euler-Maruyama with 1000 timesteps. For SGT we use the likelihood-informed inductive bias, see Section 4.4, with $k = 0$ for the preconditioner. We use $\nu = 1$ for the covariance operator in Equation (26).

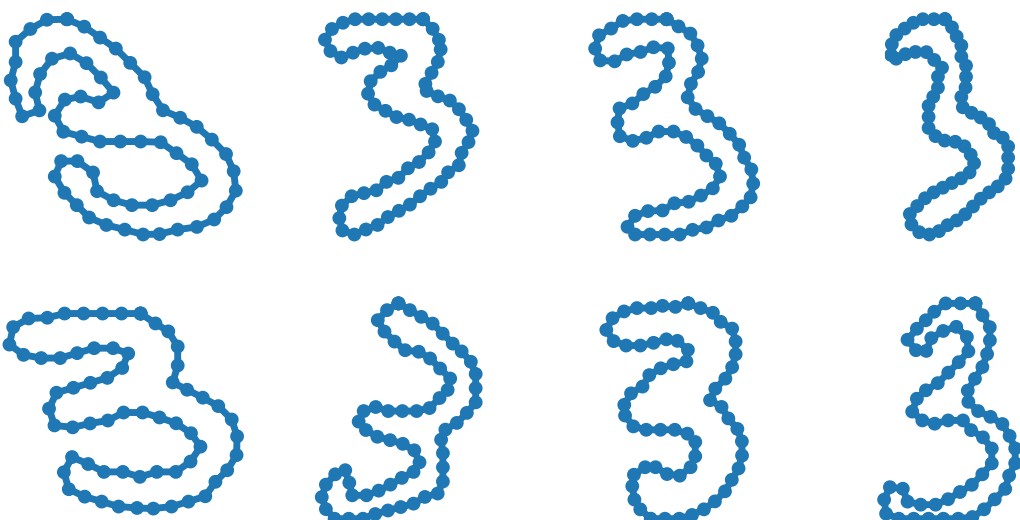

*Figure 5.* The first 8 examples of the MNIST test dataset, parametrised using 64 landmarks.

**Conditional Sampling** We compare against FunDPS (Yao et al., 2025). Note that the standard version of FunDPS trains a joint score model on both the observations and ground truth data and solves the conditional sampling task by masking one of the input channels. Here, we compare against the version in Appendix D of Yao et al. (2025), which is closer to the finite-dimensional version of DPS (Chung et al., 2023). FunDPS has a hyperparameter $\gamma > 0$ for scaling the likelihood term. We tune this value via a grid search on the first 5 shapes of the training set. Similar to observations in the finite-dimensional case, we find that the choice of $\gamma$ can be unstable. A large $\gamma$ is often required to satisfy consistency to measurements. However, for a large $\gamma$ the reverse SDE becomes unstable and sometimes diverges. The final values for $\gamma$ were, $\gamma = 10.0$ for 2 coeffs., $\gamma = 20.0$ for 4 coeffs., $\gamma = 100.0$ for 6 coeffs. and $\gamma = 165.0$ for 8 observed coefficients. We discretise the reverse SDE using Euler-Maruyama with 1000 timesteps.

### C.4. Finite-dimensional vs. Infinite-dimensional

A key advantage of the infinite-dimensional diffusion framework is its ability to generate samples at resolutions different from those used during training. In contrast, standard finite-dimensional diffusion models typically struggle to generalise across resolutions. This phenomenon was also observed in Figure 2 of Hagemann et al. (2025). Here, we reproduce a similar experiment using our datasets and models.

We train a 1D-U-Net within the finite-dimensional diffusion framework using the dataset from Section C.1, where signals are discretised on 32 grid points. Further, we train the neural operator model in the infinite-dimensional framework on the same dataset, also discretised to 32 points. We then evaluate both models by generating samples at resolutions different from the training resolution.

Although the U-Net is fully convolutional and can therefore be applied to inputs of higher resolution, this does not guarantee meaningful generalisation. To assess this, we generate samples at resolutions $n = 32$, 64, and 128 for both models. The samples are shown in Figure 7. The infinite-dimensional diffusion model generates consistent samples across resolutions with the same structural characteristics. In contrast, the finite-dimensional model fails to generalise in the same way. Samples at lower resolutions remain reasonable. However, those generated at $n = 128$ grid points exhibit significantly higher-frequency artifacts and deviate from the structure observed at the training resolution.

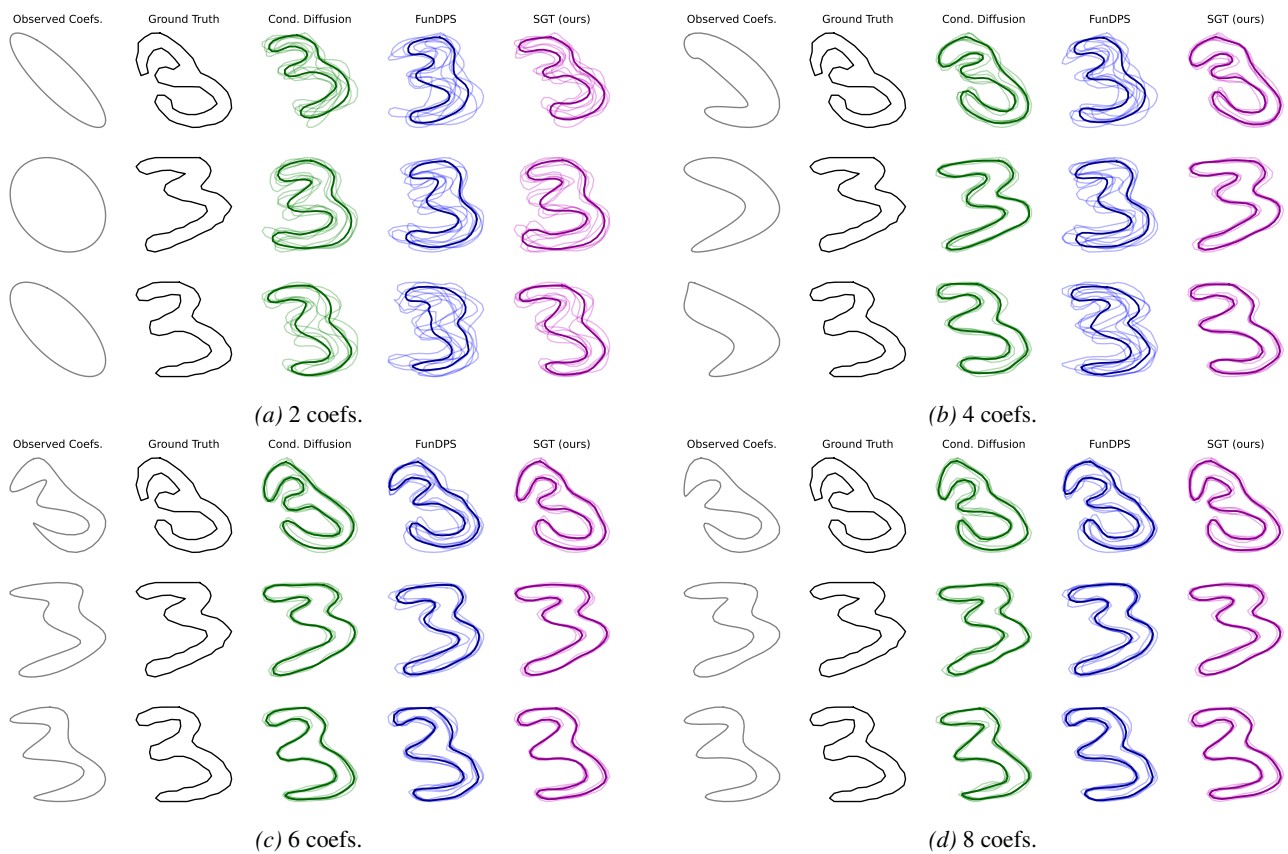

*Figure 6.* Conditional sampling results for the shape data. We show the ground truth shape and 5 samples per method (overlay). The first column is the reconstruction of the shape from the first $(2, 4, 6, 8)$ coefficients.

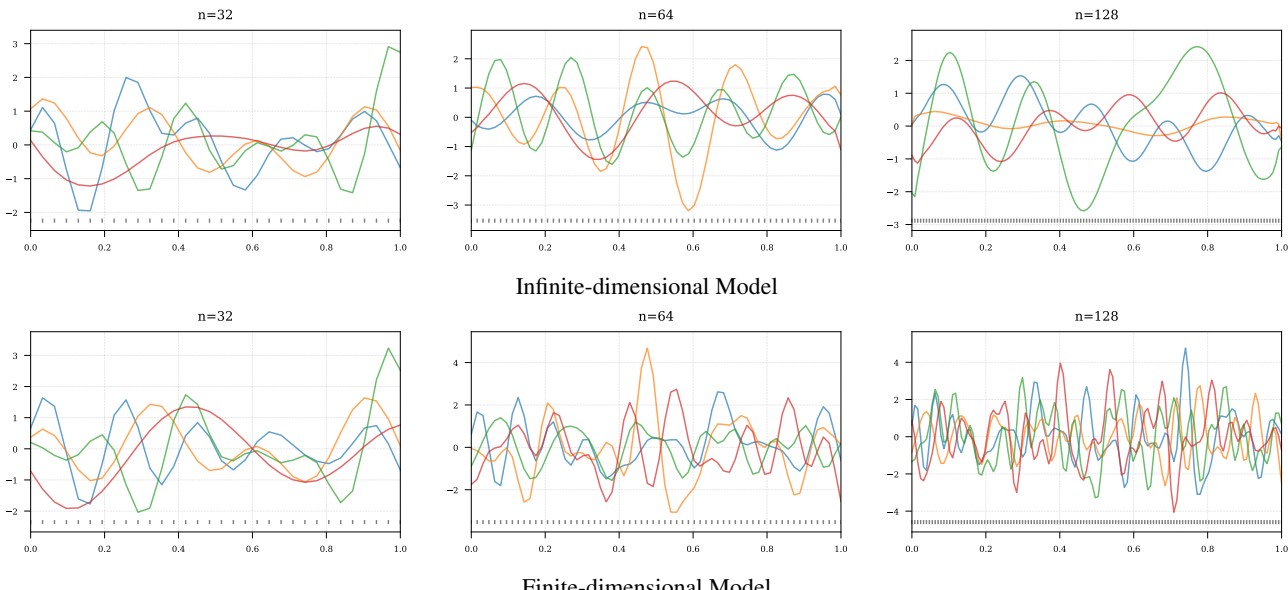

*Figure 7.* Comparison of unconditional samples from a finite-dimensional (trained at resolution $n = 32$) and an infinite-dimensional diffusion model (also trained using only resolution $n = 32$). Samples are generated at resolutions $n = 32, 64,$ and $128$.

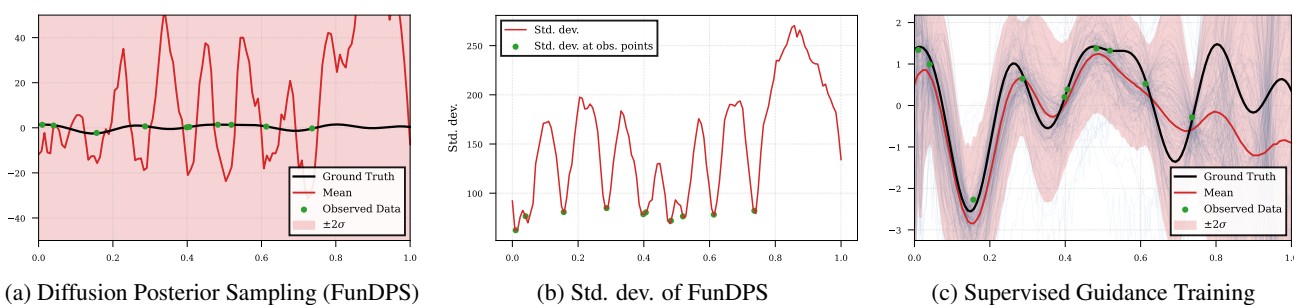

(a) Diffusion Posterior Sampling (FunDPS)   (b) Std. dev. of FunDPS   (c) Supervised Guidance Training

*Figure 8.* We compare SGT and FunDPS with a randomly initialised base model. With an untrained base model FunDPS is unstable, whereas SGT is able to produce somewhat realistic samples. The FunDPS approximation results in a high standard deviation over the full interval (see (b)).

