# OpenReview forum: "Supervised Guidance Training for Infinite-Dimensional Diffusion Models"
_ICML.cc/2026/Conference — ICML 2026 regular_

### Official Review · Reviewer_TKaZ · 2026-03-08

**Soundness:** 4
**Presentation:** 3
**Significance:** 2
**Originality:** 2
**Overall Recommendation:** 4
**Confidence:** 3

**Summary:**

This work proposes an approach to condition score-based diffusion models for sampling posterior distributions over functions using an infinite-dimensional extension of Doob's h-transform. The authors prove the existence of the conditional diffusion process, provide a decomposition of the conditional score into an un-conditional score and a guidance term, and provide a  simulation-free score matching objective to learn the guidance term. The framework is related to stochastic optimal control and other approximate methods (e.g., FunDPS) and validated on benchmark problems.

**Compliance With Llm Reviewing Policy:**

Affirmed.

**Key Questions For Authors:**

We have a few questions for the authors:
- How is $C$ determined for a pre-defined un-conditional model. Alternatively, how is the scaling parameter $\nu$ set in the covariance operator (26) for practical applications?
- It would be helpful if the authors commented why there is no benefit on the heat equation of directly encoding the smoothness of the covariance operator as compared to $k=0$? Could you clarify if $C$ misspecified in this example or known exactly based on the choice of $f$?
- How was the distribution for the training data selected for the heat equation? Was it the effect of this choice (two Gaussian bumps) on the learned guidance term?

**Limitations:**

While the approach outperformed FunDPS on some, it would be best to outline the sensitivity of the proposed approach vs. FunDPS to exactly knowing several parameters, e.g., the true covariance operator and $\gamma$ in (20). Moreover, is the proposed approach more susceptible to training variance and errors from score matching on the un-conditional score as compared to FunDPS?

**Strengths And Weaknesses:**

Soundness: The technical details were all correct and supported with appropriate theory.

Presentation: The presentation of infinite-dimensional inverse problems, stochastic optimal control and diffusions was clear, concise and self-contained for readers who are new to this topic. Our main suggestion is to avoid notation clashes, e.g., K is used for both the Hilbert space in the loss (e.g., equation 21) and the number of landmarks in section 5.4.

Significance and Originality: The work extends a line of existing work on infinite-dimensional diffusion models that has already pointed out the importance of selecting the covariance operator $C$ appropriately. Most importantly, the proposed approached is contrasted with FunDPS, and shown to outperform on several examples.

---

> ### Author Rebuttal · Authors · 2026-03-30
>
> Thanks a lot for your time and effort in reviewing our paper! We appreciate your feedback and address your review comments and questions below.
>
> - Presentation: Thanks for noting the clash of notation for $K$. We have now updated the notation for the number of landmarks to $R$.
>
>
>
> **Q1 Choice of $C$ or $\nu$:** Thank you for the question. We will add more details on this into our paper. For choosing $C$ in the unconditional model we may follow the advice of [1] Section 6.1. Here, they note that the error of the learned sample distribution depends on the Wasserstein distance between $\pi_\text{data}$ and $\mathcal{N}(0, C)$. Hence, if we know that the data distribution is absolutely continuous with respect to a Gaussian distribution $\mathcal{N}(0, \hat{C})$ (i.e. Setting 1.) then we should choose $C= \hat{C}$ to ensure that $\pi_\text{data}$ and $\mathcal{N}(0, C)$ are as similar as possible. If we do not have this information, we should instead choose $C$ so that it’s big enough that it’s Cameron-Martin space includes $\pi_\text{data}$ (so that we are in Setting 2.) and after that so that it’s as smooth as possible. That is, in both settings it is important that $C$ is chosen to either be as rough as or rougher than data samples, and after this it should be as smooth as possible: respectively, $\nu$ should be as small as necessary, and as big as possible. We also note that [1] has experiments on this (see Figure 3.) where we can see the importance of the choice of $C$.
>
> **Q2 Heat Equation:** Thanks for the question! We interpret this as being a result of the smoothness of the heat equation problem. If the score itself $\nabla_x \log h(t, x)$ is smooth, then the model must already learn to approximate a smooth function, and therefore whether it learns the smoothing operator $C$ or not does not make a big difference. For $C$ we use $Cf = \sum_{k} \frac{1}{k^{\nu}}\langle f, e_i\rangle e_i$ (as in Eq. (26)) with $\nu=0.05$. More broadly, the inverse problem for the heat equation is comparatively simple. This is reflected in the substantially lower RMSE and ES values compared to the sparse inpainting example. In this regime, architectural choices (including whether smoothness is encoded explicitly) have a smaller impact on performance.
>
>
> **Q3 Choice of data for heat equation:** We chose the data this way to satisfy the Dirichlet boundary conditions $(f(1)=f(0)=0)$.
>
>
> **Limitation 1: Sensitivity to parameters.** Both FunDPS (and other guidance-based methods) and our SGT approach rely on the likelihood gradient $\nabla \Phi$, but their sensitivities w.r.t. the choice of parameters differs. In particular, there are two covariance operators to consider:
> - Covariance in the SDE ($C$): This is a modeling choice tied to the prior and should match the unconditional model. Sensitivity of the unconditional model to $C$ is discussed in Section 7 of [1].
> - Covariance of the measurement noise ($C_\eta$): In our inverse problems, $Y=G(X_0) + \eta$ where $\eta \sim N(0, C_\eta)$. For our experiments $G$ maps to a finite-dimensional space and we choose $C_\eta = \sigma_\eta^2 I$. In FunDPS, the noise level $\sigma_\eta$ is incorporated into the parameter $\tilde{\gamma}=\gamma/\sigma_\eta^2$, which must be tuned (e.g., to minimise RMSE). In our experiments, we tuned $\tilde{\gamma}$ directly for FunDPS, but SGT does **not** require knowledge of $\gamma$ or $\sigma_\eta$. Instead, these parameters are implicitly learned via $u_\phi^2$ in Eq. 25 (the parametrisation of the guidance term).
>
> We have conducted a sensitivity analysis of the FunDPS to $\tilde{\gamma}$; see our response to Reviewer 8Dka Q4. This distinction will be clarified in the revised manuscript.
>
> **Limitation 2: Reliance on pre-trained score** This is an interesting question and we have now conducted an experiment to compare the performance of SGT v.s. FunDPS with (i) an untrained unconditional score model, (ii) a partly trained unconditional score model ($50$ epochs) and (iii) a fully trained variant ($1000$ epochs). Please refer to our response to Reviewer 8Dka (Q3) for details. We will also include this experiment into our paper!
>
> [1] Pidstrigach et al. https://arxiv.org/pdf/2302.10130

---

> > ### Author Rebuttal · Reviewer_TKaZ · 2026-04-04
> >
> > Thank you for the reply to my concerns and additional experiments! My concerns have been addressed.

---

### Official Review · Reviewer_8Dka · 2026-03-08

**Soundness:** 4
**Presentation:** 3
**Significance:** 3
**Originality:** 3
**Overall Recommendation:** 5
**Confidence:** 4

**Summary:**

The paper studies the theory of conditioning diffusions in infinite dimensional function spaces. Under two different assumptions, they provide a strong theoretical derivation, extending doobs h transform and how the conditional score decomposes. To learn the conditional part of the score they propose a practical objective and show how it performs in different numerical experiments.

**Compliance With Llm Reviewing Policy:**

Affirmed.

**Final Justification:**

My concerns have been addressed and I decide to keep my positive score

**Key Questions For Authors:**

1. Do the assumptions necessary for the theory hold for all numerical experiments?

2. Can you comment on the gains over finite dimensional approaches? In the numerical results there is a small gain over finite dimensional approaches. I would like to know how it compares in terms of computational efficiency or any other advantages that the infinite dimensional provides. Can you further comment on the resolution invariance claim that you also mention in the inpainting example?

3. Can you comment on the convergence analysis when the score of the prior is not perfectly learn?

4. How easy is it to obtain the optimal value for the scaling parameter gamma? Can you provide some results showing the sensitivity of the method to this parameter?

**Limitations:**

yes

**Strengths And Weaknesses:**

Strengths:
- The paper is very well structured and easy to follow.
- The theoretical extension of conditioning diffusions to infinite dimensional spaces is very strong. Analysing the scenario of discretising at the end while performing the analysis in the infinite dimensional setting is very interesting.
- The proposed final objective is theoretically motivated and easy to implement.


Weaknesses:
- The method relies on perfectly learn scores of the prior distribution, while in practice these are learnt from data. There is no analysis of the how this would affect convergence.
- The practical gains over finite dimensional approaches might be limited.

---

> ### Author Rebuttal · Authors · 2026-03-30
>
> Thanks a lot for taking the time to review our paper! We are happy about the positive review and the feedback. We answer your questions below:
>
> **Q1 Assumptions:** We shall try to make this clearer in the revised manuscript, but indeed the assumptions of the theory are satisfied in the experiments. For both the heat and sparse data experiment our datasets consist of functions that are sums of Gaussians or sines and cosines resulting in smooth functions that are bounded on bounded domain, and hence square integrable. Similarly, the shapes are smooth and bounded since we model them as truncated sums of Fourier basis elements. We then work in Setting 2, where $C$ must picked such that the datasets lie in the Cameron-Martin space of $C$. Since all experiments consist of smooth and square integrable functions, they trivially live in the Cameron-Martin spaces of the corresponding $C$ chosen for the experiments.
>
> **Q2 Advantage over Finite-Dim. Models:**  Thanks for the question. Please refer to our answer to Reviewer 7K8w Q2 for a discussion on the advantages of infinite-dimensional diffusion models and the resolution invariance. We also have added a section to the appendix to address this as we agree motivating the infinite-dimensional diffusion models is important for our method! Regarding in particular the computational side, the resolution invariance means that we do not have to retrain our model if we would like to use it at a higher resolution (as opposed to UNets).
>
> **Q3 Imperfect Score:** We do rely on the learned unconditional score being accurate in our method. However, we note that this is an assumption for most guidance methods. Although a theoretical convergence analysis would be interesting, this is out of scope for now. However we have instead run an extra experiment to see how SGT fares with an imperfect score compared to FunDPS which we detail below.
> We will add this to our paper as we think this is an important situation to consider.
>
> *Experiment: Imperfect unconditional score*
> We evaluate SGT and FunDPS for three different base models:
> 1) untrained base model (randomly initialised),
> 2) partly trained unconditional model ($50$ epochs) and
> 3) fully trained model ($1000$ epochs)
>
> for the sparse observation experiment. Note that 3) is the situation studied in the current manuscript. Please see the table below.
>
>
> Both FunDPS and SGT degrade under an imperfect unconditional model. This is expected as FunDPS only guides the unconditional model, whereas SGT can partly compensate for a imperfect unconditional model due to the training of the guidance term. Finally, for a random base model, FunDPS fails to produce accurate results and diverges for larger values of $\gamma$. SGT is able to partly compensate but also loses a lot of the performance. Please see https://bashify.io/i/uClBOY for conditional sampling in this setting. SGT is able to roughly recover the true function, whereas FunDPS fails to produce a realistic sample.
>
> | Sparse Observation | ES ($\downarrow$) | RMSE ($\downarrow$) |
> | -------- | -------- | -------- |
> | FunDPS (random base model, $\gamma=0.1$) |  20.685 | 558.25 |
> | FunDPS (imperfect uncond score, $\gamma=1.0$)     |3.816  | 0.472 |
> | FunDPS  (fully trained, $\gamma=1.0$)   |  3.608 | 0.447 |
> | SGT (random base model) | 6.591 | 1.238 |
> | SGT (imperfect uncond score)     | 3.230 | 0.397     |
> | SGT  (fully trained) | 3.190  | 0.392 |
>
>
> **Choice of $\gamma$ for FunDPS**
> We choose $\gamma$ to minimise the RMSE. We note that the scaling parameter $\gamma$ is only used in the DPS style approximation and is not needed for SGT. In the following table we provide results for different values of $\gamma$ for the sparse observations experiment as reported in Table 1 (where we used $\gamma=1.5$). It is known that these DPS approximations become instable for large values of $\gamma$ (also in finite dimensions, see [1]), which we also observe for $\gamma=5.0$. However for $\gamma \in [0,1]$ the results are very stable.
>
> | Metrics | $\gamma=1.0$ | $\gamma=1.5$  | $\gamma=2.0$ | $\gamma=3.5$ | $\gamma=5.0$ |
> | -------- |-------- |  -------- |-------- |-------- |-------- |
> | RMSE ($\downarrow$)    | $0.46\pm0.25$ | $0.46\pm0.26$  |  $0.46\pm0.27$  | $0.72\pm0.95$ | $21.58\pm33.19$ |
> | ES  ($\downarrow$)   | $3.79\pm1.99$ | $3.71\pm2.11$  |  $3.73\pm2.26$  | $3.78\pm2.42$ | $5.02\pm2.85$ |
>
> [1] Chung et al. https://arxiv.org/abs/2209.14687

---

> > ### Author Rebuttal · Reviewer_8Dka · 2026-04-02
> >
> > Thanks for the reply to my concerns and additional experiments! My concerns have been addressed.

---

### Official Review · Reviewer_7K8w · 2026-03-13

**Soundness:** 3
**Presentation:** 3
**Significance:** 2
**Originality:** 3
**Overall Recommendation:** 4
**Confidence:** 3

**Summary:**

This paper studies how diffusion models can be used to solve inverse problems defined over Hilbert spaces.  In particular, the authors proposed Supervised Guidance Training that conditions a pretrained diffusion model using a guidance model based on Doob's h-transform.  With the pretrained unconditional score, the guidance model can be learned from minimizing a quadratic loss similar to the denoising score matching.

**Compliance With Llm Reviewing Policy:**

Affirmed.

**Final Justification:**

My main concerns were about the fairness of the empirical comparison and the practical value of the infinite-dimensional formulation, and the rebuttal addressed both through a matched ablation and a clearer explanation of when the function-space viewpoint is advantageous. I believe the work makes a meaningful contribution and is likely to be of interest to the inverse-problems community. Overall, the rebuttal strengthened my assessment and supports a weak accept.

**Key Questions For Authors:**

1. For both the sparse-observation and heat-equation experiments, a smaller network and a less number steps are used for SGT, which is used to show the superiority of SGT.  Can the authors please do a matched ablation where the same number of parameters and steps are used? I am not fully convinced that a large number of parameters and steps are required for the conditional diffusion to achieve good results for the numerical examples presented in the paper.

2. I'm wondering if there is a discussion on when this infinite-dimensional function-space viewpoint provides advantages over discretization.

**Limitations:**

Yes

**Strengths And Weaknesses:**

Strengths:
1. The paper has a thorough theoretical justification and derivation.  It provides clear connections between formulas like reverse-time SDE and denoising score matching in the finite-dimensional and in the infinite-dimensional settings.  It has a strong justified mathematical foundation instead of a heuristic one.
2. The paper is well-motivated and the method can be useful for the inverse-problem community.

Weaknesses:
1. The numerical examples feel narrow compared to the generality of the theory.  The numerical examples are not rich enough to demonstrate the benefit of using SGT.
2. The practical advantage of this infinite-dimensional formulation is not very convincing to me considering that we can discretize the function and use finite-dimensional diffusion models.

---

> ### Author Rebuttal · Authors · 2026-03-30
>
> Thanks a lot for the positive review! We really appreciate your feedback and address your questions below.
>
> **Experiments** Our main contribution is theoretical and as such our numerical experiments are primarily designed to validate the mathematical framework rather than optimise large-scale architectures. However, we agree this is important future work (see Q2).
>
> **Q1 Machted Ablations:** We have now conducted a matched ablation where the conditional diffusion model is trained with approximately the same number of parameters and training steps as SGT. The results are reported below and will be included in the revised version of the manuscript, as we agree that it makes the motivation and advantage of SGT clearer.
>
> | Sparse Obs. | ES ($\downarrow$) | RMSE ($\downarrow$) |
> | -------- | -------- | -------- |
> | CondModel (full size) | 2.766 | 0.347 |
> | CondModel (SGT size)  | 3.244 | 0.398 |
> | SGT (k=0)       |  3.148  | 0.390  |
>
> | Heat Equation | ES ($\downarrow$) | RMSE ($\downarrow$)  |
> | -------- | -------- | -------- |
> | CondModel (full size) | 0.351 | 0.045  |
> | CondModel (SGT size)  | 0.385 |  0.049 |
> | SGT (k=0)       | 0.346 | 0.044  |
>
> We note that full SGT sampling involves both the unconditional model and the guidance model. Hence, the total number of parameters used during inference in SGT (unconditional + guidance networks) is larger than that of the conditional diffusion model. However, whereas we only need to re-train the guidance network for a new observation setting, the conditional network has to be trained from scratch.
>
>
> **Q2 Advantages of Infinite-dimensional Diffusion:**
> Thank you for the question. We agree motivating infinite-dimensional diffusion models is important for our work so we have now added an extra section that discusses the importance of infinite-dimensional diffusion models (see also [3, 4]).
>
> First, we agree that in many settings (particularly imaging applications on regular grids) the *discretise-then-learn* framework using finite-dimensional diffusion models has been very successful. However, this approach becomes more restrictive in scientific computing and PDE-based applications, where data often lives in function space and is discretised on irregular grids. As such, the advantages of the infinite-dimensional framework are as follows:
> - *Resolution invariance*: A key advantage of the infinite-dimensional formulation is resolution invariance. Prior work has shown applying fully convolutional architectures to higher resolutions often degrades in performance (see [3, Fig. 2]). In contrast, inf-dim diffusion models allow sampling at arbitrary resolutions or on entirely different meshes without retraining (see [3, Fig. 3]). This flexibility has been exploited in recent work such as FunDPS, which introduces a multi-resolution sampling pipeline. We further include a similar experiment to [3, Fig.2 and Fig.3] on our sparse observation dataset, where we observe similar resolution-generalisation behaviour (see the figure https://bashify.io/i/XmYCHV for details). This result has been added to the revised manuscript.
> - *Modeling flexibility:* The inf-dim framework offers greater modelling flexibility. In particular, it allows the use of structured covariance operators for the noise process, enabling the incorporation of prior knowledge of the data such as smoothness, correlation structure or boundary conditions.
> - *Convergence theory*: When working purely in finite dimensions, discretisations of inf-dim diffusion processes may not behave consistently under refinement. For example, consider an inf-dim SDE $dX_t = -X_t d_t + C dW_t$ with score function $s_t$, and a discretisation of $X_t$, $dX^N_t = -X^N_t d_t + C^N dW^N_t$ (e.g. by taking the first $N$ basis elements for a basis of choice) with score function $s_t^N$. It is not generally true that projecting $s_t$ onto $N$ basis elements will give $s^N_t$ (see Proof of Theorem 9 in [4]).
>
> That said, we acknowledge that fully realising the practical advantages of inf-dim diffusion models requires further architectural development. Finite-dimensional diffusion models benefitted from extensive architectural exploration and standardised implementations ([1,2]). In contrast, the inf-dim diffusion literature lacks a similarly established architecture. Existing work often makes use of neural operators architectures, e.g., Fourier Neural Operators, which still rely on regular grids for fast implementations via the FFT. There is currently no general-purpose analogue to the U-Net, working on arbitrary meshes, which we view as an important direction for future work.
>
> [1] Dhariwal and Nichol "Diffusion Models Beat GANs on Image Synthesis"
>
> [2] Karras et al. "Elucidating the Design Space of Diffusion-Based Generative Models"
>
> [3] Hagemann et al. "Multilevel Diffusion: Infinite Dimensional Score-Based Diffusion Models for
> Image Generation"
>
> [4] Pidstrigach et al. "Infinite-Dimensional Diffusion Models"

---

> > ### Author Rebuttal · Reviewer_7K8w · 2026-04-04
> >
> > The rebuttal fully resolves my concerns. The new matched ablation directly addresses the fairness issue in the empirical comparison, and the added discussion on the infinite-dimensional viewpoint clearly explains its advantages over discretize-then-learn approaches, especially for resolution transfer, irregular discretizations, and modeling flexibility.

---

### Decision · Program_Chairs · 2026-04-30

**Decision:**

Accept (regular)

**Comment:**

This paper makes a strong contribution by providing a principled infinite-dimensional treatment of conditioning in diffusion models, grounded in an extension of Doob’s h-transform and a clean decomposition of the conditional score. It further translates this theory into a practical simulation-free training objective for guidance learning. The main concerns raised by the reviewers were the fairness of the empirical comparisons and the practical relevance of the infinite-dimensional formulation. The rebuttal addressed both convincingly by providing a matched ablation and a much clearer discussion of the concrete benefits of the infinite-dimensional perspective. Reviewers consistently viewed the theoretical development as strong, well-motivated, and clearly presented, with clear relevance to inverse problems in function spaces.